



# A multi-hazard risk prioritization framework for cultural heritage assets

Giacomo Sevieri[1], Carmine Galasso[1], Dina D'Ayala[1], Richard De Jesus[2], Andres Oreta[2], Mary Earl Daryl A. Grio[3], Rhodella Ibabao[4]

[1]Department of Civil, Environmental & Geomatic Engineering, University College London, London, UK
[2]Department of Civil Engineering, De La Salle University, Manila, Philippines
[3]Department of Civil Engineering, Central Philippine University, Iloilo City, Philippines
[4]Department of Management, University of the Philippines Visayas, Iloilo City, Philippines

*Correspondence to*: Giacomo Sevieri (g.sevieri@ucl.ac.uk)

**Abstract.** Multi-hazard risk assessment of building portfolios is of primary importance in natural-hazard-prone areas, particularly for the prioritization of disaster risk reduction and resilience-enhancing strategies. In this context, cultural heritage assets require special consideration because of their high vulnerability to natural hazards - due to ageing and the type of constructions - and their strong links with communities from both an economic and a historical/sociocultural perspective. As part of the *Cultural Heritage Resilience & Sustainability to multiple Hazards* (CHeRiSH) project, funded by the UK Newton

Fund, this paper introduces a multi-hazard risk prioritisation framework specifically developed for cultural heritage assets. The proposed framework relies on a multi-level rapid-visual-survey (RVS) form for the multi-hazard data collection and risk prioritization of case-study assets. Because of the multi-level architecture of the proposed RVS form, based on three levels of refinement/information, an increasing degree of accuracy can be achieved in the estimation of structural vulnerability and, ultimately structural risk of the considered assets. At the lowest level of refinement, the collected data are used for the

computation of seismic and wind risk prioritization indices, specifically calibrated in this study for cultural heritage assets with various structural/non-structural features. The resulting indices are then combined into a unique multi-hazard risk prioritization index in which the intangible value of cultural heritage assets is also considered. This is achieved by defining a score expressing the cultural significance of the asset. The analytic hierarchy process is extensively used throughout the study to reduce the subjectivity involved in the framework, thus obtaining a simplified, yet robust, approach which can be adapted to different

building typologies. The proposed framework is applied to 25 heritage buildings in Iloilo City, Philippines, for which innovative, non-invasive techniques and tools for improved surveying have also been tested. Thermal and omnidirectional cameras have helped in the collection of structural data, together with drones for the inspection of roofs. Results of the study are presented and critically discussed, highlighting advantages and drawbacks of the use of new technologies in this field.



## 1 Introduction and motivations

Probabilistic risk assessment of building portfolios in natural hazard-prone areas is of paramount importance to define prioritization schemes for the design/implementation/optimization of disaster-risk-reduction (DRR) and resilience-enhancing strategies. This is even more important in developing countries, where most of the existing building stock has been designed/built according to obsolete codes (if any) and limited financial resources/coping capacities are available.

In this context, cultural heritage (CH) assets require special consideration because of their physical vulnerability, which has

been highlighted during recent catastrophic events (e.g., Fiorentino et al., 2018; World Bank Group, 2017), and their sociocultural value (e.g., European Commission, 2018). In fact, the lack of any hazard-resistant design (in most of the cases) and the presence of material degradation due to aging, together with the possible presence of structural modifications/local repair and/or partial/total reconstructions over time, result in high levels of vulnerability (e.g., Despotaki et al., 2018). In addition, assessing the expected losses for a given set of hazard scenarios is a complex task because of the tangible and

intangible values of CH assets (e.g., European Commission, 2018). The tangible value is mainly related to structural/architectural characteristics (direct losses), often hardly quantifiable due to the uniqueness of a given asset, and to the link with the economy of a region through cultural tourism (indirect losses). Moreover, CH has a symbolic value for a given community. The citizens' feeling of place and belonging and the sense of collective purpose are strongly linked to CH assets: their damage and partial/total collapse can have a huge impact on social cohesion, sustainable development and

psychological wellbeing. These aspects provide CH assets with an intangible value, which must be somehow considered in the risk assessment both at portfolio and building-specific level. All these issues together make the quantification of CH-asset exposure (i.e., the value at risk) a challenging task (e.g., European Commission, 2018).

An urgent need for integrating the specific features of CH assets into DRR plans has been recently highlighted by various national and international authorities across the world. One of the first published documents in this context is the report

prepared by the World Heritage Committee (UNESCO, 2008), which stated that *'most world heritage properties, particularly in developing areas of the world, do not have established policies, plans and processes for managing risk associated with potential disasters'*. In 2015 the UN General Assembly endorsed the *Sendai Framework for Disaster Risk Reduction 2015-2030* (UNISDR, 2015) which, for the first time, explicitly included CH in the overall agenda of DRR. The framework clearly recognizes culture as a key dimension of DRR, with CH specifically referred to under two priorities: (1) understanding disaster

risk; and (3) investing in DRR for resilience. However, the sector could also contribute significantly to priorities such as (4) enhancing disaster preparedness for effective response. These directions were transposed at European level through the publication of the *Action Plan on the Sendai Framework for Disaster Risk Reduction 2015-2030* (SWD, 2016), which promoted the collaboration between the public (e.g., governments) and the private sector (e.g., engineering consultancies, (re)insurance companies) for the implementation of resilience-enhancing strategies for CH assets. Following this idea, for the

first time, in 2018, an insurance company has been instructed by the Episcopal Conference in Italy (CEI) to provide a



(re)insurance policy for religious buildings from natural catastrophe risks in all 25,796 parishes of the 225 Italian dioceses, thus boosting the interest of (re)insurance companies and risk modellers in the CH-asset market (Sheehan, 2018).

Any DRR strategy, designed by governmental agencies or other stakeholders, should be based on a rational understanding of natural-hazard risks of large building stocks. However, performing detailed structural analyses for a large number of structures
is cost-ineffective because it would require high-performance computing and specific technical resources. Therefore, simplified methods for multi-hazard risk prioritization/assessment of building portfolios (e.g., FEMA P-154, 2015), framed in multi-level frameworks (e.g., Moratti et al., 2019), represent essential tools to prioritize further detailed analyses and any DRR and/or resilience-enhancing intervention. Such simplified methods should allow an analyst to also account for the intangible value of CH assets and to consider their specific construction features by just using a small amount of information - to be
typically collected in highly-complex urban settings, such as in developing countries.

This paper addresses the above-mentioned issues by proposing a multi-level, multi-hazard risk assessment framework for CH assets. The proposed framework relies on an ad-hoc rapid-visual-survey (RVS) form which can be used to gather information for different levels of analysis varying in refinement. At the lowest refinement level, the focus of this paper, it allows calculating risk prioritization indices against various natural hazards. Specifically, seismic and wind risk prioritization indices
for CH assets are proposed. They represent an extension of those developed within the *Indonesia School Programme to Increase Resilience* (INSPIRE; Gentile et al., 2019) and the *Safer Communities through Safer Schools* (SCOSSO; Nassirpour et al., 2018) projects respectively. In particular, the INSPIRE seismic risk prioritization index is extended to the case of unreinforced masonry (URM) buildings by providing specific performance modifiers (*Section 3.2*) and calibrating their relative weights. In a similar way, the SCOSSO wind risk prioritization index is adapted for the specific characteristics of CH-asset
roofs (*Section 3.3*). A simplified approach for the combination of the two indices, and which allows for an explicit consideration of the intangible value of CH assets (reflecting the CH-asset significance; Kerr, 2013), is also proposed (*Sections 3.4 and 3.5*). Weights and scores used in this study are calibrated through the analytical hierarchy process (AHP; Saaty, 1980) in order to reduce the subjectivity involved in the framework.

The effectiveness of the proposed framework has been demonstrated during a field survey of 25 CH assets in Iloilo City,
Philippines. With a population of 447,992 inhabitants and a 1.02% population annual growth rate, Iloilo City is one of the most highly-urbanized cities of the south-eastern tip of Panay island in the Philippines (Philippine Statistics Authority, 2016). It is also the capital city of the province of Iloilo and an important heritage hub for tourism in the Philippines. The historic street Calle Real, located in the old downtown district of Iloilo City, is home to several fine examples of historic luxury buildings constructed in the first half of the 20th century during the American colonization (ICCHCC, 2010). Most of them
have been surveyed during the fieldwork. Being located in a cyclonic region with the West Panay fault (the nearest one) just 15 km away (Yu and Oreta 2014), Iloilo City represents a perfect case study to test the proposed multi-hazard risk and resilience assessment framework.

The overall framework has been developed within the *Cultural Heritage Resilience & Sustainability to multiple Hazards* (CHeRiSH) project, funded by the UK Newton Fund, which aims to define a multi-level risk and resilience assessment





framework for CH assets in the Philippines exposed to multiple natural hazards. It also investigates innovative, non-invasive techniques and tools for CH assets survey/diagnostic as well as different retrofitting approaches for Filipino CH assets, which meet conservation and adaptive reuse criteria.

## 2 Review of risk prioritization schemes for CH assets

A number of methodologies for the vulnerability/risk prioritization of buildings are available in the scientific literature and in

international guidelines. These approaches often rely on the definition of pre-determined building classes (e.g., Lagomarsino and Giovinazzi, 2006) and corresponding fragility/vulnerability relationships for each class; alternatively, RVS forms and empirically calibrated vulnerability/risk indices based on the RVS results (e.g., Uva et al., 2016) are used. Although a comprehensive review of the current state-of-the-art in the field is outside the scope of this paper, a brief overview of relevant risk prioritization procedures defined for CH assets is presented in this section.

Even though the procedure introduced by the Federal Emergency Management Agency (FEMA P-154; FEMA, 2015) is not specifically tailored for CH assets, it represents an important reference for every risk prioritization framework based on RVS form, like the one proposed in this study. Starting from a sidewalk screening of the surveyed building, the procedure described in the FEMA P-154 document consists of 1) definition of the building type (or class) by identifying the primary gravity load-carrying material of construction and the primary seismic force-resisting system; and 2) identification of building attributes

modifying  the expected seismic performance with respect to an 'average' archetype building representative of the class. Scores can be associated to the above features, thus determining a seismic vulnerability index without performing any structural analyses. The scoring framework is directly linked to the probability of collapse of archetype buildings (FEMA P-155; FEMA, 2015) through the Hazard United States (HAZUS) model (Kircher et al., 2006).

Lagomarsino (2006) proposed one of the first multi-level frameworks for the seismic prioritization of CH assets based on the

estimation of the structural vulnerability. At the lowest refinement level, the approach allows for the computation of a vulnerability prioritization index based on a macro-seismic model (i.e., which makes use of vulnerability curves obtained through damage-assessment data collected after earthquakes of different intensities) to be used with macro-seismic intensity hazard maps. The computation of the index requires various (expert) opinions on geometrical and structural features of the surveyed building, which are then used to determine an average vulnerability index and vulnerability modifiers. At the highest

refinement level, a structural model (e.g., equivalent-frame model) is used to calculate numerical fragility curves for selected damage states. Finally, these results are used to determine (probabilistic) distributions of damage states (Lagomarsino and Giovinazzi, 2006) to assess the structural vulnerability, thus increasing the accuracy of the result. In this procedure the CH-asset value is not directly considered.

D´Ayala et al. (2006) proposed a conceptual approach for the multi-hazard vulnerability assessment of historic buildings. The

methodology is based on three steps: 1) hazard screening for the identification of the relative damageability of a given historic building; 2) selection of those hazards that can lead to damage scenarios and estimation of the expected losses through a



process of building disassembly; 3) structural analyses of important building components in order to achieve a higher level of accuracy. For each hazard, the prioritization index is defined as a holistic score obtained by using a weighted summation of scores related to the building features (e.g., structural materials, preservation condition, geometry). Besides being one of the

first multi-hazard vulnerability prioritization schemes, the study presented a comprehensive approach for assessing the tangible and intangible value of CH assets. In particular, significance and restorability of CH assets are used as reference criteria. The significance is defined essentially as a function of the authenticity and originality of the CH asset, i.e. of its historic and aesthetic character. Its evaluation is based on a wide range of criteria including social, cultural and economic attributes. Whereas, the evaluation of the restorability requires a decision making relative to possible interventions and successful

outcomes. In addition to cultural and architectural criteria (e.g., acceptability of restoration), the restorability of a damaged building depends on objective factors, such as availability of original building materials, information on the original structural features and substantial financial support. Finally, indices related to different hazards are combined by using normalized losses of common building typologies in the region with reference to a particular peril as weights.

Yu and Oreta (2015) presented a multi-hazard risk prioritization scheme for CH buildings which explicitly considered the asset

value. The risk prioritization index is defined as the weighted summation of mitigation and vulnerability factors, whose relative importance is considered through the use of the AHP for the calculation of the weights. The authors proposed an innovative procedure for the quantification of the tangible and intangible value of CH assets based on both objective and subjective criteria. The asset value is determined by "Cultural Heritage" factors, such as architectural and historical values, and "Economic/Tourism" factors, such as commercial use, tourism importance and adaptive reuse adaptability. The total asset

value is given by the weighted summations of all these characteristics, where the weights are calibrated through the AHP and based on expert judgments. The scores related to each characteristic are derived through a "focus group discussion" consisting of different stakeholders, such as technicians, historians and inhabitants.

D'Ayala et al. (2016) proposed a procedure for the multi-hazard vulnerability prioritization and assessment of CH assets based on structural models and synthetic scores related to information gathered in a specifically-defined RVS form. In particular, the

Failure Mechanisms Identification and Vulnerability Evaluation (FaMIVE) method (D'ayala, 2005) is used to calculate the seismic vulnerability and then a seismic prioritization index. An engineering-based load and resistance approach, which considers both pullout failure of the first fastener (screw or nail) and pullover failure of the first roof panel, is used to assess the wind vulnerability. Structural components and system resistances (i.e., capacity) are treated as uncertain parameters in the simulations, while gravity and wind load effects (i.e., demand) are considered deterministic (Song, et al. 2019). The CH asset

value is considered only in the assessment of the flood vulnerability, which is based on RVS form and it defines the prioritization index as the average of scores related to different vulnerability factors.

Despotaki et al. (2018) presented a procedure for the evaluation of the seismic risk of CH sites in Europe for prioritization purposes. The approach exploits the methodology proposed by Lagomarsino (2006), discussed above, for the calculation of baseline vulnerability indices. In order to consider the uniqueness of each asset, vulnerability indices are adjusted based on

specific parameters of monuments (e.g., position, state of maintenance or the damage level). The authors applied the proposed



procedure to important UNESCO (United Nations Educational, Scientific and Cultural Organization) sites, thus highlighting its feasibility in the vulnerability assessment of large CH building portfolio.

Moratti et al. (2019) proposed a multi-level approach for the seismic assessment of URM churches based on five levels of data collection which lead to three levels of analysis refinement. At each level, performance indices are calculated as ratio of the

structural capacity and the seismic demand, both expressed in terms of displacement. At the lowest refinement level, statistical data of church characteristics, which not require building inspections, are used to perform displacement-based assessments in which structures are approximated through single-degree of freedom (SDoF) systems. The second refinement level requires building inspections in order to define SDoF models for each pier constituting the surveyed churches. In this way, the same methodology developed for the lowest refinement level can be applied also in this case. The highest refinement level requires

detailed data in order to build proper global in-plane structural models and local out-of-plane models. The global seismic behaviour can be evaluated by using SDoF models of each pier or multi-degree of freedom (MDoF) models (e.g., equivalent-frame models), which are then used within displacement-based assessment methods in order to apply the same procedure defined for the previous levels. The local out-of-plane behaviour is assessed through kinematic analyses, linear or non-linear one.

This brief literature review shows that the few prioritization approaches which explicitly consider the tangible and intangible value of a CH asset and/or multiple hazards require detailed information about the structure under investigation, since they are based on an explicit loss estimation exercise. This can contrast with the nature of prioritization methods at portfolio scale which should require only small amount of data. Moreover, as discussed in *Section 1*, such procedures are widely used in developing countries where specific data are usually not available, this requiring several simplifying assumptions. The

quantification of losses for CH assets is further complicated by the subjective definition of the asset intangible value and the difficulties in assigning a value to their non-market nature.

## 3 The CHeRiSH framework for the multi-hazard risk prioritization of cultural heritage assets

As discussed above, the multi-hazard risk prioritization approach proposed in this study is part of a broader project (CHeRiSH) which has different objectives involving civil and structural engineering as well as social science, arts and humanities. From

the engineering perspective, the project aims at investigating innovative, non-invasive techniques and tools for CH assets survey and diagnostic, and to develop new methods/models, and their implementation tools for the multi-hazard risk and resilience assessment of CH assets. The main focus of the project is on the exposure and physical vulnerability modelling of CH assets as well as on the prioritization of resilience-improving solutions for selected assets through multi-criteria decision making. Whereas, from the social science perspective, the main objectives are related to the promotion of community

awareness on the vulnerability of CH assets and the design of disaster risk communication and emergency management campaigns targeted at cultural organizations and local communities.





The overall risk and resilience assessment framework proposed in CHeRiSH has a multi-level structure (Figure 1), consisting of three refinement levels which are directly linked to the amount of available information. The lowest refinement level allows for a risk prioritization of the CH portfolio, while the others two levels can allow for the estimation of the structural vulnerability, and ultimately structural risk at building-specific scale, thus increasing the accuracy of the analysis.

Specifically, the multi-hazard risk prioritization procedure for CH assets (lowest refinement level) proposed in CHeRiSH can be seen as a five-step procedure, only requiring few basic information about the structures under investigation. These five steps are: 1) data collection through a sidewalk survey (by means of the proposed RVS form); 2) selection of the hazard-intensity level (e.g., for a selected mean return period) for which the prioritization is needed; 3) calculation of risk prioritization indices for different hazards; 4) combination of the different single-hazard prioritization risk indices; and 5) calculation of multi-hazard risk prioritization indices which accounts for CH asset intangible values, and building ranking.

At the second refinement level, data from both the interior and exterior are used to build simplified structural models which allow improving the assessment of the structural performances. Since no specific information about materials is available at this refinement level, the parameters of the structural models are treated as random variables or assumed based on simulated design. At the highest refinement level structural drawings are required to develop detailed structural models for the evaluation of the CH asset performance for various loading conditions. Material test results can also be used for the calibration of numerical models, thus reducing the uncertainty of the results.

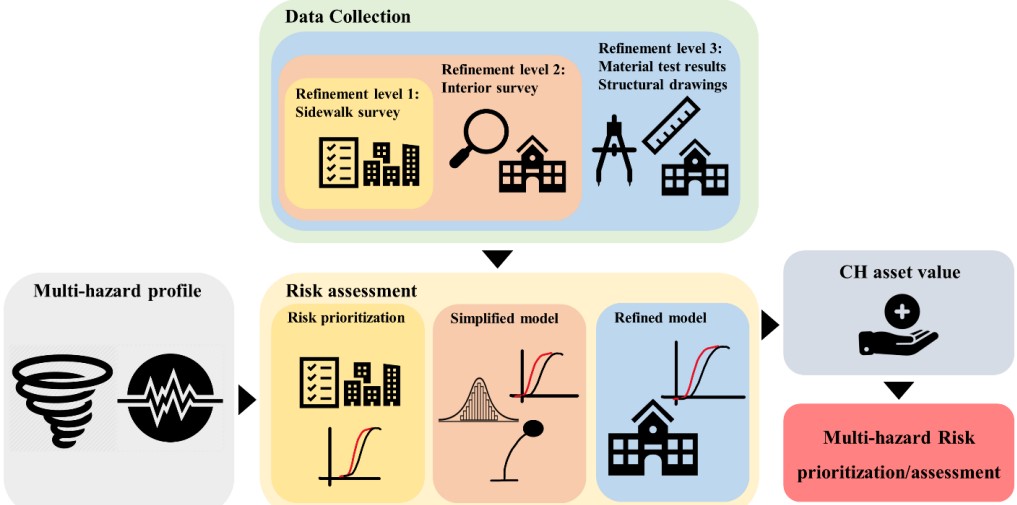

**Figure 1: CHeRiSH Multi-level, multi-hazard risk assessment framework.**

## 3.1 The CHeRiSH Rapid Visual Survey form

The proposed RVS form has been designed in order to account for the specific features of Filipino CH assets, which mainly consist of reinforced concrete (RC) frames and masonry or mixed structures. In fact, according to the Filipino Republic Act no. 10066 (2009), also known as the *National Cultural Heritage Act*, the only "objective" feature which defines a building as



a CH asset is the year of construction. Structures which are at least fifty years old can be declared to be a "Heritage House" by
the National Historical Commission of the Philippines (NHCP). Differently from the criteria applied by UNESCO (2017) for
the definition of CH assets, the Filipino law does not explicitly consider subjective features of the buildings such as the
architectonical value and sociocultural factors. Therefore, fairly recent RC frame-type structures, characterized by limited
architectural and/or cultural features, are often part of the Filipino CH portfolio. Considering these specific characteristics of
the Filipino CH assets, the proposed RVS form has been designed for various structural typologies employing different
construction materials and lateral-load resisting systems.

As discussed above, the proposed RVS form (Figure 2) is defined in a multi-level framework. The basic information required
for the first level of refinement can be collected by means of a sidewalk survey of the building by trained engineers in
approximately 20-30 minutes, depending on the size of the construction. The second level of refinement/accuracy (light grey
entries) requires more detailed data on the structure (e.g., presence of non-continuous structural walls, type and quality of roof-
to-wall connections, diaphragm typology, among many others) which can be collected only by surveying the building both
from its exterior and interior. The third level of refinement/accuracy (dark grey entries) requires material test results and
structural drawings in order to calibrate reliable numerical models.

The RVS form is composed of six sections over three pages; it includes various parts related to the general identification and
geolocation of the building, its geometric properties (including space for sketching the building's shape and footprint), and its
structural characteristics and deficiencies, including the structural typology and the dimensions/details of the main structural
members. It is also possible to assign a "Confidence Level" for each parameter, thus accounting for the degree of uncertainty
in the collected data. Special emphasis has been placed on the design of "Vulnerability Factors" and the "Roof Information"
sections. The "Vulnerability Factors" section contains a list of vulnerabilities which can be found in the survey of masonry or
RC structures. In addition, CH assets in the Philippines are particularly vulnerable to typhoon-induced strong wind, as recent
catastrophic events have demonstrated. Since the main collapse mechanisms due to extreme wind and typhoons are related to
the failure of roofs (Vickery et al., 2006), the "Roof Information" section requires data about the roof geometry, its structure
and connection to the walls, the quality and the conservation of the materials and fasteners. The data collected in the CHeRiSH
RVS form are fully compatible with both the Global Earthquake Model (GEM) building taxonomy (Brzev et al., 2013) and
the HAZUS model. Hence, existing prioritization indices based on these two models can also be used within the CHeRiSH
framework.



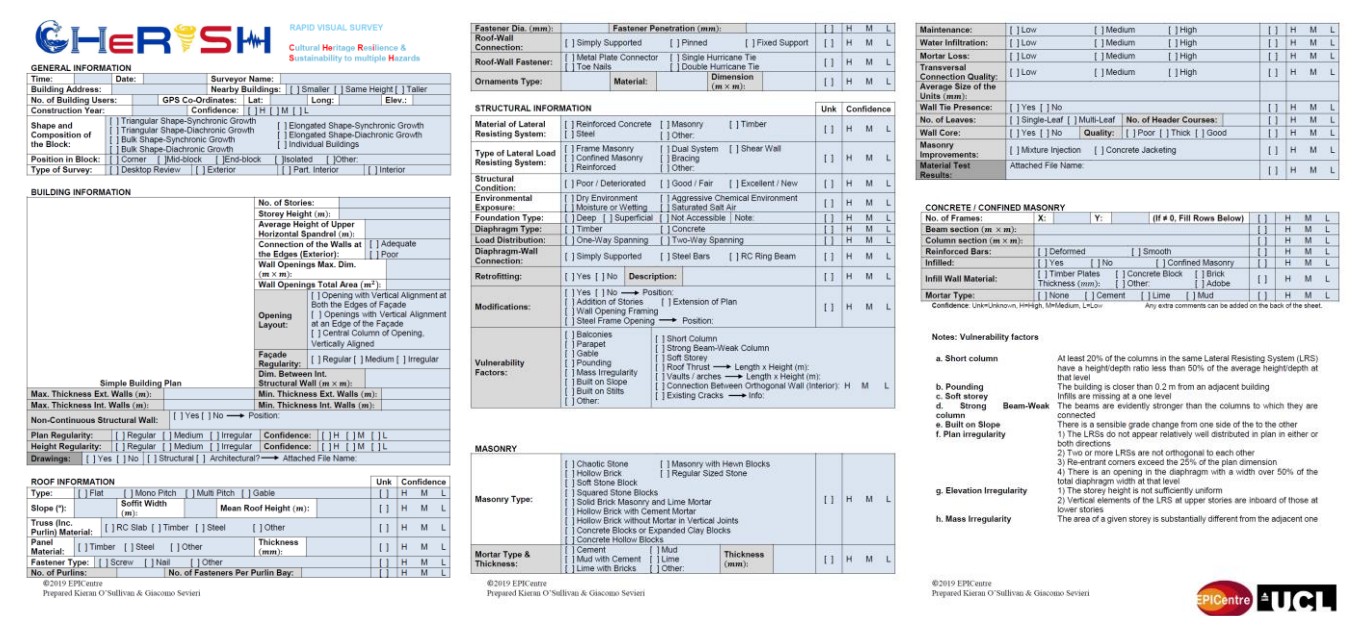

**Figure 2: CHeRiSH RVS form.**

### 3.1.1 The use of new technologies for CH assets survey and diagnostic

CH assets located in highly-populated cities are deeply integrated within the urban fabric and they may host private and public activities. This complicates and slows down survey campaigns because it limits the possibility to access areas of the construction and to properly collect data. Moreover, the time available to carry out the survey is usually limited because of the high costs involved per person-hour. In order to improve the amount and quality of the data collected without increasing the number of personnel involved, new technologies should be utilised during fieldworks.

Indeed, one of the objectives of the CHeRiSH project was to test the feasibility of applying new technologies for the survey of CH assets. In particular, omnidirectional cameras, thermal cameras, drones, photogrammetry and Building Information Modeling (BIM) have been extensively used during the fieldwork discussed in *Section 4* of this paper.

Omni-directional cameras (also known as 360° cameras) are devices that have two wide angle (> 85°) fisheye lenses mounted back-to-back, facing in opposite directions that each are able to photograph 180° of a scene. The camera can then produce two

unstitched 180° pictures which can also be stitched together to form one 360° (equirectangular) picture. 360° pictures can be used during a desktop review to build 3D point clouds of the asset interior, to find lost data and to assess the presence, type, and location of non-structural elements. Interior 3D point clouds can be used to determine distances and heights of the structural members which cannot be directly acquired in the field because of the activities hosted by the surveyed buildings. Non-structural elements can be a source of vulnerability, so their presence must be considered during the definition of resilience-

enhancing strategies.



Similarly, the collection of reliable measurements of the building exterior is a challenging task, especially in densely populated cities. Indeed, car traffic, people and temporary obstacles prevent the architectural survey. Therefore, as in the case of interior measurements, exterior point clouds can be analysed during a desktop review, allowing a more accurate definition of the building dimensions. Exterior point clouds can be built by using photogrammetry technology (e.g., Aicardi et al., 2018) which allows transforming pictures, such as the ones taken by smartphones, into measurable objects.

The quality and typology of the masonry characterizing a given asset, and the diaphragm characteristics (e.g., its orientation) are essential data needed even at the first refinement level of the proposed framework. Due to the activities hosted by the considered CH assets and their architectural value, specific (invasive) inspection tests cannot be performed. Non-invasive techniques such as thermal cameras may play an important role for the collection of this information. Thermal cameras allow one to detect infrared energy (heat) and converting it into an electronic signal, which is then processed to produce a thermal image. Since heat sensed by a thermal camera can be very precisely measured and materials are characterized by different thermal properties (e.g., emissivity coefficients), their presence within the structure can be easily detected by just taking a picture. However, the use of thermal cameras is strictly related to the presence of thermal flux within the surveyed structural element. If the system is in thermal equilibrium, the different thermal characteristics of the materials are not highlighted and then their presence cannot be detected.

The use of a quadcopter drone is a personnel multiplier and can overcome building access issues that are frequently encountered on site. Because of the unique vantage point that they offer, drones can have the most influential impact in the quality and quantity of data collected for the roof survey. It is worth noting that post-event surveys in the Philippines and around the world reveals that most economic loss in high wind-hazard areas are related to the breach of the building envelope. The breach of a building envelope typically includes roof panel uplift, roof-to-wall connection failure, roof system damage, and rupture of window and door glasses due to excessive pressure or missile impact. With the roof heavily damaged or removed, walls may become unstable without sufficient lateral support and can collapse. Hence, during strong typhoons, nonengineered roofs built with low quality materials (typical of CH assets) and showing heaving material degradation (due to aging) are highly vulnerable to wind uplift and are the main concern here. The collection of data on roof characteristic is usually very difficult because of their inaccessibility. The data required for the calculation of the wind prioritization index defined in the current study can be assessed quicker with use of a drone rather than through direct visualisation by accessing the building. The use of drones is then particularly useful to carry out a reliable roof inspection and build accurate numerical models for wind fragility estimation.

The use of new technologies, as described above, drastically increases the stream and amount of data/information which can become prohibitive to manage. Therefore, a suitable BIM platform is currently under development within the CHeRiSH project. The platform is designed to store all the data collected during the fieldwork in Iloilo City, and it will allow the creation of 3D models (architectural and structural ones) of the surveyed buildings. This can be achieved by exploiting the interior and exterior point clouds created respectively by using the photogrammetry and omnidirectional cameras. The BIM platform can also play a crucial role to access the vulnerability data of the surveyed CH assets and to manage resilience-enhancing strategies.



### 3.2 The seismic prioritization index

In this study, the INSPIRE index (Gentile et al., 2019) for the seismic risk prioritization of RC constructions is extended to URM buildings. The need for this extension is justified by the composition of the Filipino CH portfolio, which counts different structural typologies, including URM building. The INSPIRE index, and then the proposed one for CH assets ($I_S$), is an empirical proxy for the relative seismic risk of various buildings within a given building portfolio. It consists of two components: a baseline score ($I_{BL}$) and a performance modifier ($\Delta I_{PM}$), which are finally summed up to obtain the total seismic

risk index (Eq.1).

$$I_S = I_{BL} + \Delta I_{PM}, \tag{1}$$

The extension of the INSPIRE index to include URM buildings has required the definition of a proper performance modifier, as described in detail in this section. However, guidance on the computation of the RC-building performance modifier is also provided, because of the high occurrence of this structural typology within the analysed CH portfolio (*Section 4*).

The calculation of the baseline score is based on the fragility curves available in the HAZUS model (Kircher et al., 2006), which represent an harmonized and transparent framework for the multi-hazard fragility/vulnerability/risk assessment of a wide range of structures. The use of the HAZUS model as a starting point for the definition of proposed seismic risk prioritization index is further justified by the fact that several countries around the world, including the Philippines, have adopted seismic provisions which are consistent with the recommendations of the Uniform Building Code 1994 (UBC, ICBO,

1994). In fact, this code is used as a benchmark to define four seismic *code levels* in the HAZUS framework. The four *code levels* are: high, moderate, low and pre-code (not seismically designed) level. The first three levels are defined with regard to the provisions in UBC (ICBO, 1994) for seismic zone 4, 2b and 1, respectively. Indeed, the National Structural Code of the Philippines (NSCP, 2015) is the primary design code in the country, providing guidance to civil and structural engineers on the design and assessment of buildings, and any other structures since its 1st edition in 1972. Table 1 below shows the history of the NSCP. The post-2001 NSCP versions are all based on the 1997 UBC, and earlier versions were similarly based on

previous editions of the UBC, as shown in the Table 1, allowing the proposed mapping with the HAZUS code levels. Based on the data collected during the survey, four separate vintages can be identified: post-2001 (which includes also post-2010, i.e., all the building designed consistently with the UBC 1997), 1991–2001, 1971-1970, and Pre-1970 (Table 2). In this case the analysis of the results from the onsite surveys, shows that the construction practice does not seem to closely follow the design plans and specifications, the code compliance for each design vintage can downgraded by one level for the analysis.

The HAZUS fragility curves express the seismic performance of archetype buildings which are classified based on four parameters: material (*Mat*), basic structural system (*BSS*), building *Height* and seismic *Code Level*. Such fragility curves are log-normal cumulative distribution functions (CDFs) expressing the conditional probability that the given structure will reach or exceed a pre-defined damage state (DS) given the hazard intensity measure (IM). The HAZUS-model fragility curves are

defined in terms of median ($\mu$) and dispersion ($\beta$) parameters for different IMs, including the peak ground acceleration (PGA), and various DSs, i.e., slight, moderate, extensive and complete damage (see Kircher et al., 2006 for details).





**Table 1: Evolution of Seismic Codes in the Philippines**

| Philippines Design Code (Edition) | Basis for general and earthquake loading provisions |
|---|---|
| NBCP 1972 (1st edition; 2nd printing in 1977) National Building Code of the Philippines | UBC 1970 |
| NBCP 1982 (2nd edition) | UBC 1978 |
| NSCP 1987 (3rd edition) National Structural Code of the Philippines | UBC 1985 |
| NSCP 1992 (4th edition, Volume 1 – Buildings, Towers, and Other Vertical Structures; Volume 2 for Bridges published in 1997) | UBC 1988 |
| NSCP 2001 (5th edition, Volume 1 – Buildings, Towers, and Other Vertical Structures) | UBC 1997 - inclusion of Active Fault Maps from PHIVOLCS |
| NSCP 2010 (6th edition, Volume 1 – Buildings, Towers, and Other Vertical Structures) | UBC 1997 - inclusion of Active Fault Maps from PHIVOLCS |
| NSCP 2015 (7th edition, Volume 1 – Buildings, Towers, and Other Vertical Structures) | UBC 1997 - updated Active Fault Maps presented by region |

**Table 2: HAZUS Building Seismic Design Level Classifications**

| Construction data | FEMA HAZUS code compliance assignment |
|---|---|
| Post-2001 | Moderate code (for NSCP 2001 – 2010) |
| 1991–2001 | Low code (for NSCP 1992) |
| 1970-1990 | Pre-code (for NSCP 1972 – 1987) |
| Pre-1970 | No Code |


The calculation of the baseline score requires the selection of a target DS, a set of building classes (characterized by a combination of *Mat*, *BSS*, *Height* and *Code Level*), and one or more hazard levels (in terms of the considered IM). Such hazard level must be selected based on the seismicity of the considered building portfolio/geographic area and the considered performance objective. The DS exceeding probability for each considered building class can thus be computed for the

considered IM level(s). Specifically, considering PGA as the reference IM, the building basic parameters are mapped into the exceeding probability of the selected DS conditional to the PGA value, as in Eq. 2.

$$P_{HAZUS} = P(DS \geq DS_3 | Mat, BSS, Code\ Level, Height, PGA) \qquad (2)$$

Baseline scores are then calculated in order to be proportional to such exceeding probabilities after a rescaling in the range [1 %, 50 %] based on the minimum and maximum DS exceeding probability in the complete (non-filtered) HAZUS database, as

follows:

$$I_{BL} = \left(\frac{50-1}{P_{HAZUS,max} - P_{HAZUS,min}}\right)\left(P_{HAZUS} - P_{HAZUS,min}\right) + 1. \qquad (3)$$

In Eq. 3, $P_{HAZUS,max}$ and $P_{HAZUS,min}$ are the maximum and minimum DS exceeding probability in the HAZUS database for the selected level(s) of PGA, while $P_{HAZUS}$ is the DS exceeding probability of the considered building, for the chosen level of PGA. Figure 3 shows the fragility curve set related to the Extensive Damage state for RC and URM buildings adopted in this

study. The Extensive Damage state is mainly related to the life safety performance objective, but other DSs can be key to



ensure the integrity of CH assets. The aim of the study is to assess the validity of the prioritization framework in the worst-case scenario, this justifies the choice of the Extensive Damage state.

The performance modifier ($\Delta I_{PM}$) represents the perturbation of the baseline score due to the presence of vulnerability factors. Its calculation requires the definition of secondary parameters selected with respect to the construction features of the

investigated portfolio in order to complement the information in the HAZUS fragility curves. Therefore, the baseline score provides the (conditional) seismic risk of a given building class, while the secondary parameters are related to building-specific vulnerability factors.

In its original version (Gentile et al., 2019), the performance modifier is defined as the weighted summation of scores ($SCORE_{seismic}$) which describe different alternatives of each secondary parameter and which are defined on a uniform

partitioning of the range [0%, 100%], typically based on engineering judgement. The weights ($w_{SP}$) are needed to reflect the relative importance of the considered secondary parameters, which affect the seismic behaviour of buildings in different ways. In this work, the AHP (Saaty, 1980) is used to calibrate such weights. This process allows an analyst to have a rational and mathematically consistent assignment of the weights: starting from expert judgements on every possible pairwise comparison of the secondary parameters, collected into a so-called decision matrix, the AHP allows one to obtain the values of the weights

by solving an eigenvalues problem.

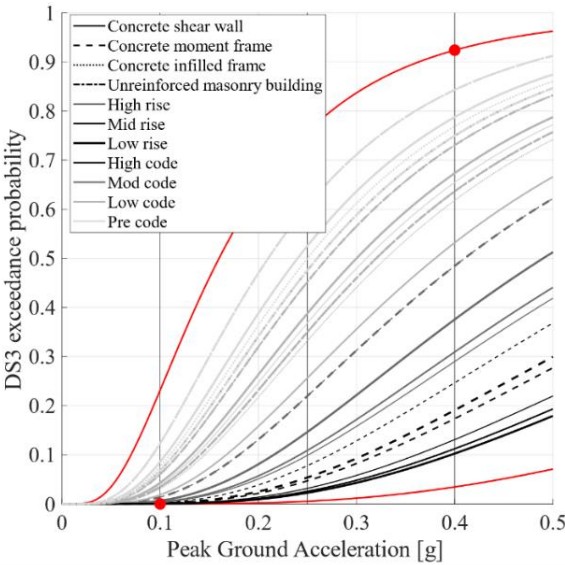

**Figure 3:HAZUS fragility curve database related to the Extensive Damage limit state for RC and URM buildings.**

In particular, the seismic vulnerability assessment of URM buildings requires consideration of the quality of the material (e.g.,

Borri et al., 2015), the out-of-plane local mechanisms (e.g., Sorrentino et al., 2017) and global (in-plane) behaviour (e.g., Lagomarsino et al., 2013). These factors, together with the presence of façade ornaments, have been considered as macro-categories for the definition of the URM-building performance modifier. According to the scientific literature (e.g., Borri et



al., 2015), the *Material Quality*, which expresses the quality of the masonry, strongly affects the seismic response of the structure. The *Material Quality* is thus calculated based on the *Masonry Typology* (e.g., Chaotic stones, Solid brick masonry

with lime mortar, Concrete blocks) and the *Masonry Degradation*. If the *Material Quality* is not sufficiently high, the structure cannot develop the so-called out-of-plane local mechanisms. Therefore, this parameter must be considered more important than the others.  The *Local Behaviour* is the second most important macro-category. Indeed, if out-of-plane local mechanisms are not avoided, the structure cannot behave as a unique fabric (e.g., Sorrentino et al., 2017). When the material quality is sufficient and the out-of-plane local mechanisms prevented, then the *Global Behaviour* must be assessed (e.g., Lagomarsino

et al., 2013) and of course it is more important than the presence of non-structural *Façade Ornaments* (Figure 3). The expert judgments (Table A.1) used in this study for the calibration of the macro-category weights ($w_{MC,m}$) through the AHP reflect these considerations. Clearly, the decision matrix adopted in this study reflects the characteristics of the Filipino CH assets and the expert opinion of the authors (academic and professional engineers across the UK and the Philippines); it should be calibrated before the entire procedure can be applied for the analysis of different building portfolio.

The secondary parameters collected within each macro-category have been selected based on the fundamental rules of masonry structure design (e.g., Heyman, 2014; Paulay and Priestley, 1992) and the commonly observed post-earthquake damage on URM structures (e.g., Fiorentino et al., 2018). For this reason, parameters related to the geometry and the regularity of the façade (*Opening Layout*, *Wall Slenderness*, *Façade Regularity* and *Opening Area*) as well as those related to connections (*Wall-to-Wall connection*, *Wall-to-Diaphragm connection* and *Wall-to-Roof connection*) are considered for the definition of

the *Local Behaviour*. Indeed, it is well known that the activation of out-of-plane local mechanisms is strictly linked to the geometry of the piers, which is also determined by the position of the openings, and the connection with orthogonal walls, diaphragms and roof (D'Ayala, 2005). In this study, the presence/quality of connections has been valued more important than the geometry/regularity of the facades, as shown in Table A.2. This is due to the fact that the Filipino CH portfolio is characterised by buildings with regular opening layouts but various diaphragm typologies, so a proper prioritization scheme

can be achieved by using the proposed judgments.

The regularity of the building (*Plane Shape* and *Storey Height Uniformity*) and the presence of vulnerability factors (*Added Storeys*, *Pounding* and *Unfavourable Soil*) are used to quantify the *Global Behaviour* of URM buildings. The regularity of the Filipino CH assets leads to assign greater importance to vulnerability factors, such as *Pounding* and *Unfavourable Soil*, rather than the others thus achieving a relatively more accurate prioritization scheme (Table A.3).

Table 3 provides guidance on the selection of the alternatives for the calculation of the URM building performance modifier. The performance modifier can be finally calculated as in Eq. 4,

$$\Delta I_{PM} = \frac{1}{2} \sum_{m=1}^{M} w_{MC,m} \sum_{n=1}^{N_m} w_{SP,n} SCORE_{seismic;m,n}, \qquad (4)$$

where $M$ is the total number of macro-categories, $N_m$ is the number of secondary parameters within the $m$-th macro-category and the subscript $n$ indicates the considered secondary parameter.





The secondary parameters for the calculation of the RC structure performance modifier are selected according to Gentile et al. (2019). Having no macro-categories in this case, the weights $w_{MC,m}$ in Eq. 4 are assumed equal to 1, while the secondary parameters weights $w_{SP,n}$ are calibrated through the AHP to reflect the expert judgments indicated in Table A.4; see Gentile et al. (2019) for a critical discussion on the assumptions made here. These parameters express the *Preservation Condition* of the material, the regularity of the structure (*Plane Shape*, *Storey Height Uniformity* and *Added Storeys*), the presence of

vulnerability factors (*Infills at Ground Storey*, *Short Column* and *Pounding*) and the soil conditions (*Unfavourable Soil*).

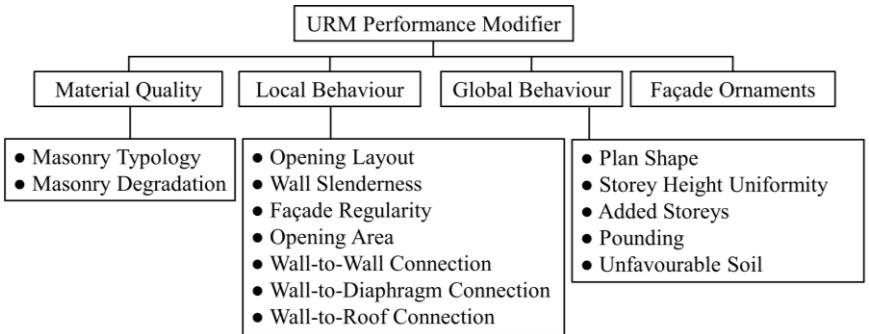

**Figure 4: Performance modifier scheme.**

The expert judgments expressing the relative importance of the considered RC-building secondary parameters (Table A.4) are

calibrated accounting for the peculiarities of Filipino CH assets. In particular, infills at ground storey, short column and pounding have been valued more important than the other secondary parameters. Indeed, many Filipino CH assets have non-engineered structures resulting from reconstructions and/or modifications over time. Therefore, these three vulnerability factors are commonly diffused. This choice results in a higher variability in the prioritization scheme. Table 4 provides guidance on the selection of the alternatives for the assignation of scores to the secondary parameters.

One of the most important advantages of the proposed approach is the possibility to easily adapt it for the prioritization of other building typologies by simply considering various secondary parameters and modifying the expert judgments a to reflect different construction features and their relative importance on the asset vulnerability. Only the consistency of the opinions must be checked through the calculation of the consistency index ($CI$) as in Eq. 5, after the pairwise comparison:

$$CI = \frac{\lambda_{max} - r}{r - 1} \qquad (5)$$

In Eq. 5, $\lambda_{max}$ is the largest eigenvalue, calculated as solution of the AHP, while $r$ is the rank of the judgment matrix. Finally, the $CI$ is compared to the random consistency index ($RCI$), which is the average consistency index of a large number of randomly generated reciprocal matrices. If the $CI$ is smaller than 10% of the $RCI$, the final values of the weights are logically sound and not a result of a random prioritisation. When such a criterion is not satisfied, the whole process should be repeated until an acceptable consistency is achieved (Saaty, 1980). The consistency condition is satisfied for all the comparisons used





in the definition of the seismic index (Macro-categories: $CI = 0.0477 \leq 0.09 = 10\% RCI$; Local behaviour: $CI = 0.0246 \leq 0.132 = 10\% RCI$; Global behaviour: $CI = 0.0615 \leq 0.112 = 10\% RCI$).

**Table 3: Macro-categories and secondary parameters for URM buildings: definition, alternatives, scores and weights.**

| Macro-category | $w_{MC}$ | Secondary Parameters | $w_{SP}$ | Alternatives | Scores |
|---|---|---|---|---|---|
| Material Quality | 0.4607 | Material Typology | 0.5 | Chaotic stones | 100 |
| | | | | Hollow brick / Regular sized stone | 50 |
| | | | | Solid brick masonry and lime mortar / Concrete blocks | 0 |
| | | Material Degradation | 0.5 | Significantly affecting performance (Poor structural condition) | 100 |
| | | | | Moderately affecting performance (Good structural condition) | 50 |
| | | | | Not affecting performance (Excellent structural condition) | 0 |
| Local Behaviour | 0.2894 | Opening Layout | 0.0582 | Opening with vert. alignment at both edges of the façade | 100 |
| | | | | Opening with vert. alignment at only one edge of the façade | 50 |
| | | | | Opening with vert. alignment at the centre of the façade | 0 |
| | | Wall Slenderness | 0.0346 | High ($h/l \geq 10$) * | 100 |
| | | | | Medium ($5 \leq h/l \leq 10$) | 50 |
| | | | | Low ($h/l \leq 5$) | 0 |
| | | Façade Regularity | 0.0975 | Irregular (openings are not aligned) | 100 |
| | | | | Medium (openings are vertically aligned) | 50 |
| | | | | Regular (openings are horizontally and vertically aligned) | 0 |
| | | Opening Area | 0.0468 | High (more than 50% of the total façade area) | 100 |
| | | | | Medium (between 25% and 50% of the total façade area) | 50 |
| | | | | Low (less 25% of the total façade area) | 0 |
| | | Wall-to-Wall Connection | 0.1923 | Poor | 100 |
| | | | | Adequate (mechanical connection) | 0 |
| | | Wall-to-Diaphragm Connection | 0.3696 | Poor | 100 |
| | | | | Adequate (ring beam) | 0 |
| | | Wall-to-Roof Connection | 0.2010 | Poor | 100 |
| | | | | Adequate (mechanical connection) | 0 |
| Global Behaviour | 0.1901 | Plan Shape | 0.1732 | L-shape or irregular | 100 |
| | | | | C-shape | 50 |
| | | | | Rectangular or regular | 0 |
| | | Storey Height Uniformity | 0.1125 | Significantly non-uniform (more than 0.5m difference) | 100 |
| | | | | Moderately non-uniform (difference between 0 and 0.5 m) | 50 |
| | | | | Uniform | 0 |
| | | Added Storeys | 0.1021 | Yes | 100 |
| | | | | No | 0 |
| | | Pounding | 0.4307 | Pronounced (less than 0.1m gap) | 100 |
| | | | | Moderate (gap between 0.1m and 0.2m) | 50 |
| | | | | None (more than 0.2m gap) | 0 |





| | | | | |
|---|---|---|---|---|
| | Unfavourable Soil | 0.1815 | Yes (very soft soil; liquefaction is not explicitly considered) | 100 |
| | | | No | 0 |
| Façade Ornaments | 0.0598 | | Yes | 100 |
| | | | No | 0 |

* $h$ and $l$ are the wall height and thickness respectively.


**Table 4: Secondary parameters of RC buildings: definition, alternatives, scores and weights.**

| Secondary Parameters | $w_{SP}$ | Alternatives | Scores |
|---|---|---|---|
| Preservation condition and/or existing damage | 0.0939 | Significantly affecting performance (Poor structural condition) | 100 |
| | | Moderately affecting performance (Good structural condition) | 50 |
| | | Not affecting performance (Excellent structural condition) | 0 |
| Plan Shape | 0.0826 | L-shape or irregular | 100 |
| | | C-shape | 50 |
| | | Rectangular or regular | 0 |
| Storey Height Uniformity | 0.0470 | Significantly non-uniform (more than 0.5m difference) | 100 |
| | | Moderately non-uniform (difference between 0 and 0.5 m) | 50 |
| | | Uniform | 0 |
| Added Storeys | 0.0470 | Yes | 100 |
| | | No | 0 |
| Infills at ground storey | 0.3039 | Yes | 100 |
| | | No | 0 |
| Short column | 0.1817 | Yes | 100 |
| | | No | 0 |
| Pounding | 0.1817 | Pronounced (less than 0.1m gap) | 100 |
| | | Moderate (gap between 0.1m and 0.2m) | 50 |
| | | None (more than 0.2m gap) | 0 |
| Unfavourable Soil | 0.0621 | Yes (very soft soil; liquefaction is not explicitly considered) | 100 |
| | | No | 0 |

### 3.3 The wind prioritization index

The proposed wind prioritization index for CH assets ($I_W$) is based on the vulnerability factors proposed by Nassirpour et al.

(2018) for the definition of the SCOSSO index, a multi-hazard vulnerability prioritization index for Filipino schools. The authors proposed a scoring method based on ratings related to specific building features which are combined to determine an overall damageability index. Particularly important for the aims of this study is the set of roof vulnerability factors related to the wind hazard. The authors considered eight construction features, also used in this study, which represent: the entire building construction features (*Code level* and *Number of Storeys*), the roof construction features (*Roof Structure*, *Roof Covering* and

*Roof Pitch*) the *Roof Connection*, and the material conditions (*Roof Condition* and *Structural Condition*). As for the case of the seismic prioritization index, the code level follows the classification proposed by the HAZUS model (Kircher et al., 2006). Adopting the same code classification for the seismic and wind indices enables the proposed procedure to be consistent.


The proposed wind prioritization index ($I_W$) is defined as a proxy for the relative wind risk of the considered buildings within the analysed portfolio. In fact, $I_W$ (Eq. 6) is calculated as the weighted summation of scores ($SCORE_{wind}$) related to the
structure of the roof and the presence of vulnerability factors (Table 5), which are then multiplied by a hazard parameter ($\widehat{w}_H$).

$$I_W = \widehat{w}_H \sum_{i=1}^{8} w_{VF,i} SCORE_{wind,i} \tag{6}$$

The score values are in the range $[0\%, 100\%]$ and they allow analysts to convert a qualitative judgment on the status of a particular vulnerability factor into a quantitative indicator. The hazard parameter reflects the wind hazard of the region where the analysed asset is located. Even though the wind hazard in the Philippines is fairly homogeneous, three regions are herein
considered: west coastal areas (low wind hazard), central part of the country (medium wind hazard) and east coastal regions (high wind hazard). In fact, according to the National Structural Code of the Philippines (2015), the wind hazard increases from the east coast to the west coast of the country.

The combination weights ($w_{VF,i}$) are calibrated through the use of AHP to reflect their relative importance, according to the expert judgments reported in Table A.5. As discussed in the previous sections, the non-engineered nature of the Filipino CH
asset roofs promotes pullout (fastener) and pullover failures (panel). Therefore, the *Roof Connection* is considered the most important parameter. Immediately after that, material conditions and *Construction years* play a fundamental role. Degraded materials can lead to the roof failure even if good quality connections are installed, while modern constructions should ensure a higher level of reliability than older ones (given good connections and materials). The remaining parameters can affect the roof system behaviour only if those previously listed are negligible. The judgments assumed for the wind vulnerability factors
in this application lead to $CI = 0.0297$ and $RCI = 1.41$, thus satisficing the consistency condition.

The AHP is also used to calibrate the values of the hazard parameters ($\widehat{w}_H$), reflecting the judgment matrix reported in Table A.6. Clearly, areas with high wind hazard are valued more important than medium and low wind hazard. The hazard parameters ($\widehat{w}_H$) are finally determined by normalising the AHP weights ($w_H$) as shown in Table 6. The consistency index and the random consistency index are $CI = 0.046$ and $RCI = 0.58$ respectively.

**3.4 Combination of risk prioritization indices**

Once prioritization indices related to different hazards are calculated, they must be properly combined in order to obtain a comprehensive indicator of the relative multi-hazard risk of the assets within the analysed portfolio.

In this study the multi-hazard risk prioritization index ($I_{multi}$) is calculated as the Euclidian norm of the vectors whose components are the $k$ single-hazard prioritization indices ($I_k$) (Eq. 7).

$$I_{multi} = \sqrt{\sum_k I_k^2} \tag{7}$$

Eq. 7 can be applied only if the single-hazard risk prioritization indices ($I_k$) have the same range of variation. However, the resulting multi-hazard risk prioritization index ($I_{multi}$) will be characterised by a different range. This can be rescaled in any other desired range without affecting the prioritisation list of the considered building portfolio. This simple combination rule





does not introduce any further subjectivity into the framework, and it can be applied even when numerous hazards are
considered. However, this method does not consider neither the interaction of different hazards at the various levels of the risk
assessment chain nor weights for the different hazard prioritization indices.

**Table 5: Wind vulnerability factors: definition, alternatives, scores and weights.**

| Vulnerability Factors | $w_{VF}$ | Alternatives | Scores |
|---|---|---|---|
| Code level | 0.1623 | Pre-code | 100 |
| | | Low code | 66 |
| | | Moderate code | 33 |
| | | High code | 0 |
| Number of storeys | 0.0436 | More than 3 storeys | 100 |
| | | 2:3 storeys | 50 |
| | | 1 storey | 0 |
| Structural condition | 0.1725 | Deteriorated / poor | 100 |
| | | Fair / good | 50 |
| | | New / excellent | 0 |
| Roof Structure | 0.0838 | Bricks | 100 |
| | | Timber truss | 66 |
| | | RC slab | 33 |
| | | Steel truss | 0 |
| Roof Covering | 0.0671 | Tiles | 100 |
| | | Iron sheets | 50 |
| Roof Pitch | 0.0943 | Multi-pitch | 100 |
| | | Mono-pitch | 50 |
| | | Flat | 0 |
| Roof Condition | 0.1715 | Deteriorated / poor | 100 |
| | | Fair / good | 50 |
| | | New / excellent | 0 |
| Roof Connection | 0.2049 | Deteriorated / poor | 100 |
| | | Fair / good | 50 |
| | | New / excellent | 0 |

**Table 6: Wind hazard parameters.**

| Wind hazard | $w_H$ | $\widehat{w}_H$ | Description |
|---|---|---|---|
| High hazard | 0.540 | 1 | East coastal areas (basic wind speed with a 15% probability of exceedance in 50 years: between 290 kph and 320 kph). |
| Medium hazard | 0.297 | 0.550 | Central part of the country (basic wind speed with a 15% probability of exceedance in 50 years: between 270 kph and 290 kph). |
| Low hazard | 0.163 | 0.302 | West coastal areas (basic wind speed with a 15% probability of exceedance in 50 years: between 240 kph and 270 kph). |

Loss curves (i.e., loss values versus their annual probability of exceedance) for various individual hazards, and calculated for
a specific region, show different non-linear trends (Fleming et al., 2016). Therefore, considering different return periods, the
relative effect of two catastrophic events (related to two different hazards) on the built environment may completely change.
For instance, for low return periods, such as 100 years, earthquake and extreme-wind economic losses are comparable, while
for high return periods, such as 1000 years, the economic loss related to seismic events is usually higher than that related to





extreme-winds. This fact may be considered within the proposed framework by defining suitable combination weights for the single-hazard prioritization indices in Eq. 7. Such combination weights should vary with the mean return period of interest selected for the prioritization in order to express how every considered hazard contribute to the total loss. This would require
a priori loss curves, which are usually not available for developing countries.

**3.5 The value of CH assets**

The proper definition of the asset exposure is a fundamental step of the risk assessment process, requiring the quantification of the asset value. As discussed in *Sections 1* and *2*, this task is particularly complex for CH assets because of their multiple impacts (e.g. economic, social, spiritual) which cannot be solely determined in monetary terms, similarly to other building
typologies. Moreover, the relatively broad definition of cultural heritage adopted in different countries (no standardised definition exists; e.g., European Commission, 2018; Filipino Republic Act no. 10066, 2009) makes even more complex the quantification of the CH asset exposure. Most of the methods proposed in the scientific literature neglect the CH asset exposure, thus considering vulnerability prioritization indices or assuming a homogeneous exposure for the whole building portfolio.

The simplified approach for considering the intangible value of CH assets in the prioritization scheme (lowest refinement
level) proposed in this study assumes that the tangible values (direct and indirect costs) is constant for the entire portfolio, so that it does not affect the prioritization scheme.  As discussed in *Section 1*, the intangible value is peculiar to each specific CH asset, and then it cannot be considered constant for the entire portfolio. Therefore, a score approach is proposed for its quantification through the calculation of the CH value index ($I_{CH\ value}$). It assumes the intangible value linked to the significance as "monument" of the CH asset by adopting the classification issued by Kerr (2013). Four categories are
considered for the definition of the scores: *Word Heritage*, *National Heritage*, *National/Local Heritage* and *Local Heritage*. Table A.7. shows the expert judgments assigned to express the relative importance of each significance category and needed for the calculation of the scores through the AHP. The judgments express the idea that the intangible value increases with the significance of the analysed CH asset. Table 7 provides guidance for the selection of the appropriated CH significance and it reports the relative scores for which the consistency condition is satisfied ($CI = 0.01\ \leq RCI = 0.9$).
Finally, after a normalization process of the CH value index ($I_{CH\ value}$), which allows for the calculation of $\hat{I}_{CH\ value}$, the multi-hazard risk prioritization index which considers the CH value ($I_{multi,CH\ value}$) can be calculated as

$$I_{multi,CH\ value} = I_{multi}\hat{I}_{CH\ value}.\qquad\qquad(8)$$

**Table 7: CH significance scores.**

| CH status | $I_{CH\ value}$ | $\hat{I}_{CH\ value}$ | Description |
|---|---|---|---|
| Exceptional significance | 0.4673 | 1 | The CH asset is considered a world heritage; it is characterised by an exceptional significance recognised worldwide. |
| Considerable significance | 0.2772 | 0.5932 | The CH asset is listed among the CH assets of national interest; it has national significance and it is possibly protected by national organisations. |
| Some significance | 0.1601 | 0.3426 | The CH asset has features of national significance but insufficient to be recognised as CH of national interest. |





| Little significance | 0.0954 | 0.2042 | The CH asset is characterised by local significance, so it has no national significance. |
|---|---|---|---|

# 4 Case-study: CH assets in Iloilo City, Philippines

## 4.1 Description of Filipino CH assets

Recent catastrophic events, e.g., the M7.2 2013 Bohol earthquake or the 2013 Typhoon Haiyan, have highlighted how Filipino CH assets are particularly vulnerable to natural hazards due to ageing and type of construction. As already discussed, CH assets and communities are doubly tied because of their economic and social connections. This link is even more important in developing countries where the cultural tourism is seen as one of the priority sectors by which governments aim to foster inclusive and sustainable socio-economic development, due to its potential for job creation and revenues. For instance, according to the Philippines Statistics Authority (2019) the contribution of tourism to the Philippine economy was 12.7 % of GDP in 2018.

The proposed multi-hazard framework for risk prioritization of CH assets has been tested on 25 CH buildings located in Iloilo City, Philippines (Figure 5), one of the oldest cities and a touristic hub in the country, which contains a collection of historic sites, monuments, and CH buildings. Realizing the importance of preserving its heritage, the city government has actively pursued the advocacy of promoting the city's culture, by identifying heritage zones and instituting a Heritage Conservation Council to oversee and promote CH preservation.

With three active faults in the near proximity of the city, Iloilo City is listed under Seismic Zone 4 in the official seismic map of the Philippines by the Philippine Institute of Volcanology and Seismology (National Structural Code of the Philippines, 2015). According to GEM (Pagani et al., 2018), the seismic hazard in Iloilo City, in terms of PGA with a 10% of probability of exceedance in 50 years, is in the range 0.35g to 0.55g. Since the city is also situated in Zone II of the Philippines Wind Zone Map (i.e., the three-second gust speed at 10m above the ground is equal to 117 km/h by assuming a return period of 50 years), it represents a perfect case study to assess the feasibility of the proposed approach.



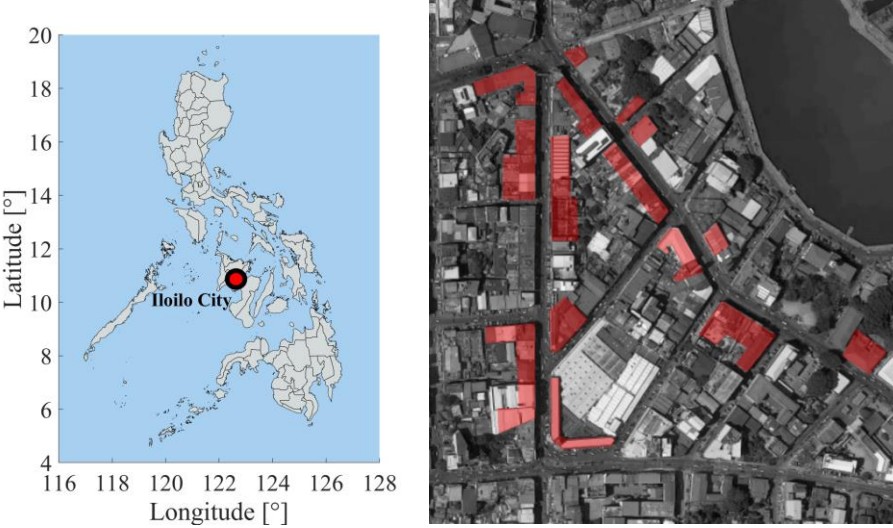

**Figure 5: Surveyed CH buildings in Iloilo city, Philippines. Background imagery by ©2019 CNES / Airbus, Maxar Technologies, map data by ©2019 Google.**

The analysed building portfolio is composed of URM and RC frame-type structures. Most of the building construction years are dated around the beginning of the last century; however, during their operational life, the Iloilo City CH assets experienced catastrophic events (e.g., earthquake and fire) which led to their partial or total reconstruction. As discussed above, new technologies have been used during the fieldwork in order to help the surveyors in the data collection exercise. In particular, drones have been extensively used for façade and roof inspections. As an example, Figure 6a shows the façade of the "Villanueva building" (ICCHCC, 2010), while Figure 6b shows the building roof. The "Villanueva building" is a L-shape, two-story RC frame, whose roof was inaccessible; the drone was the only practicable tool for collecting roof data/information. The only limitation on the use of drones was the strong wind during the fieldwork, which strongly affected the flight capability. This important aspect must be considered when a survey campaign has to be organized in a cyclonic region. Figures 6c and Figure 6d respectively show the "Villanueva building 6" (ICCHCC, 2010) façade and its point cloud obtained by elaborating the pictures taken by smartphone and photo camera. Photogrammetry is a powerful tool for the construction of point clouds, but specific practical rules must be followed to obtain good quality results. This technology requires high quality pictures of the façades with a specific overlapping, according to the software used during the elaboration step. A good quality point cloud can be obtained only if the façade is clear enough of obstacles, such as cars and people. This aspect must be considered during the planning phase of the survey campaign. Ideally, the pictures needed for photogrammetry should be taken during the hours in which there is less traffic, usually early morning.




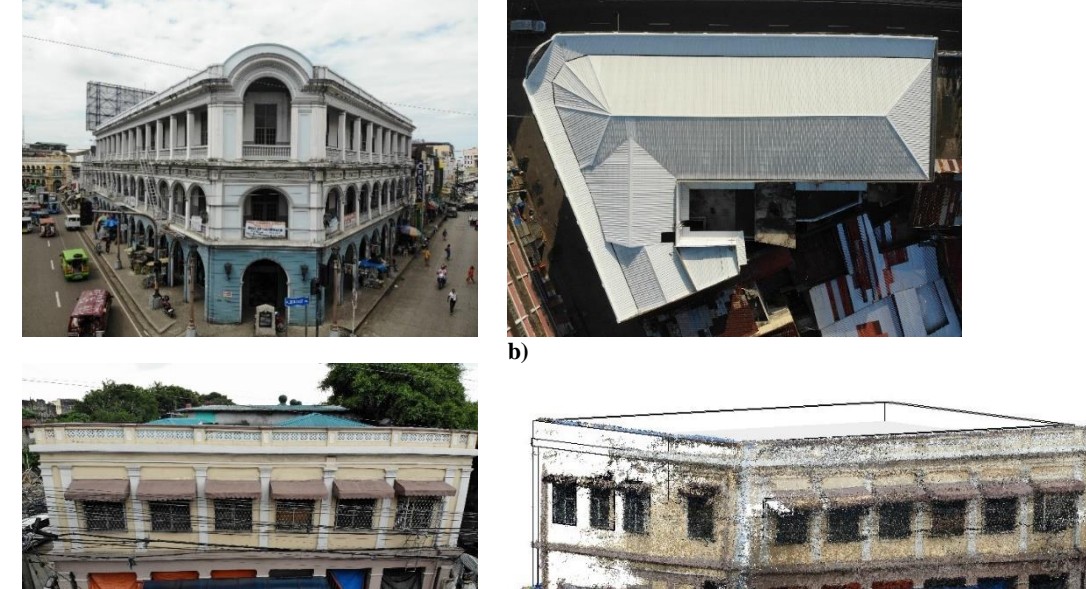

**Figure 6: Use of new technologies for the survey of the Iloilo City CH assets: Villanueva building front façade (a), and roof (b) by drone; Villanueva building 6 frontal façade (c) and point cloud (d) by drone and photogrammetry respectively.**

### 4.2 Main statistics of the data collected during the fieldwork

The main statistics derived from the data collected during the fieldwork are reported in Figure 7. Most of the surveyed CH assets are two-story (Figure 7a), plan-regular buildings (Figure7b), somehow justifying their good performance during the M7.8 1948 Lady Caycay earthquake, the second largest event in the 500-year history of Philippine seismic activities (Geoscience Australia, 2012). The surveyed buildings are located within a complex urban context; in fact, they are parts of blocks with different shapes and compositions (Figure 7c), thus complicating the estimation of their seismic vulnerability. The statistics of the *Structural condition* (Figure 7d) highlight the level of degradation and the lack of maintenance for the assets under investigation. Specifically, 60% of the surveyed buildings show *Structural conditions* which moderately affect the building performances. This means presence of deficiencies which may moderately affect the structural performance, such as small cracks concentrated on a limited number of structural elements and infill panels, and/or limited damage of the roof. Whereas, 36% of the considered assets shows *Structural conditions* which may significantly affect the building performance, such as widespread cracks on structural elements, concrete cover crushing with rusty rebars and extended damage of the roof. Most of the structure deficiencies are due to a poor quality of the construction materials. The unusually large dimension of the aggregates together with an extreme heterogeneity in their distribution within the structural elements are the main causes of the bad performance of the materials.



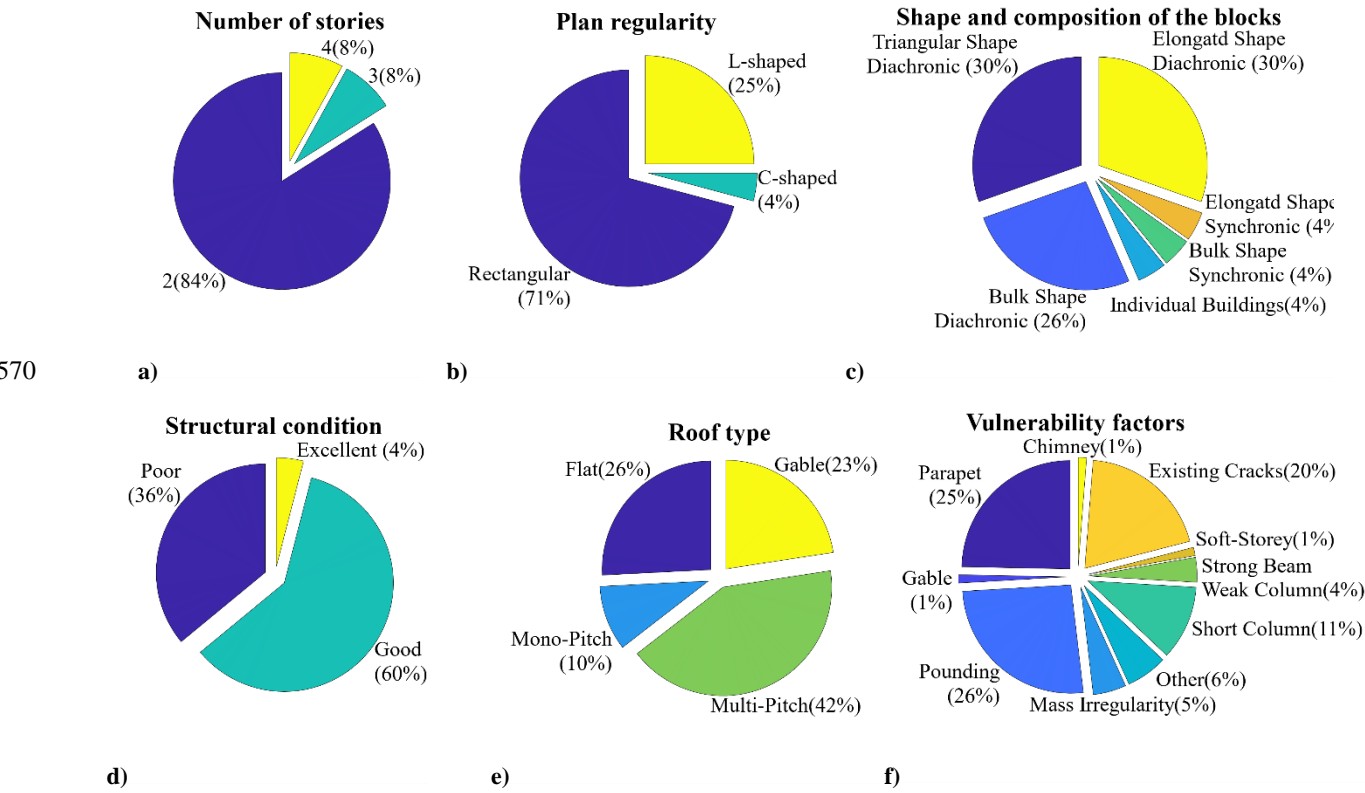

a)                              b)                                        c)

        d)                              e)                                        f)

**Figure 7: Statistics for the 25 surveyed CH buildings, Iloilo City, Philippines.**

Figure 7f shows a widespread presence of various vulnerability factors. The most common and dangerous vulnerability is the

potential for pounding and the presence of short columns. This can be explained by the use of obsolete codes during the design

and construction of these assets. Moreover, regarding the potential for pounding, the high annual population growth rate in

Iloilo City has led to construction in all the available space, without concern for the distance between buildings. According to

Figure 7e, various typologies of roof made by different construction materials can be found. Flat roofs are mainly made by

concrete, while gable, mono- and multi-pitch ones are generally characterised by a timber structure and metal roof sheets. An

advanced degradation level affects the elements of the roofs, the structure and also the connections, i.e. fasteners and roof-to-

wall connections, thus further increasing their vulnerability.

### 4.3 Prioritization scheme

The collected data have been finally used for the calculation of the risk prioritization indices proposed in this study (*Section

3*). The resulting indices are arbitrarily categorized in three groups, respectively "green, yellow and red tags" by defining two

thresholds. The definition of such thresholds is essentially a subjective (often political) choice that shapes the prioritization

scheme, based for instance on resources availability. For a governmental agency, those can be calibrated estimating the average





structural retrofit (or relocation) cost per building and defining the amount of available public funding in two or more-time windows (e.g. one and five years) to obtain specified DRR objectives. As a proof of concept, in this paper the thresholds are selected to be equal to 33% and 66% for the calculated seismic, wind or multi-hazard indices.

The seismic risk prioritization indices (Figure 8a) show fairly homogeneous baseline scores, indicated with grey bars. This is due to the common construction features of the analysed CH assets. In fact, most of them are regular RC frame structures built before the 1970, and so they are considered pre-code structures. Figure 8a also highlights how important the performance modifiers, and so the vulnerability factors, are in the definition of the seismic prioritization scheme. The analysed CH assets have common vulnerability factors, in particular *Pounding*, and diffused degradation. These increase the values of the seismic

risk prioritization indices, in fact only four assets are below the 33th percentile. This also leads to a relatively small variability of the results. Due to relatively small extension of the survey area, the same *Unfavourable Soil* condition are assumed for all CH assets (Table 3).

The wind risk prioritization indices (Figure 8b) show a higher variability if compared with the seismic ones. This is mainly due to the different construction features and degradation conditions of CH asset roofs observed during the survey. Highly

degraded roofs are strongly penalised by the scores considered in this study (Table 5). Therefore, structures with the worst maintenance conditions show the highest values of the wind risk prioritization indices. In this study, all of the CH assets are considered located in the same hazard region (medium hazard Table 6).

The two indices are finally combined following the procedure proposed in *Section 3.4* thus obtaining the multi-hazard prioritization indices ($I_{multi}$) shown in Figure 8c. The results clearly indicate that the wind hazard plays a substantial role in

determining the prioritisation scheme for the CH assets in Iloilo city. Indeed, the overall trend of the multi-hazard results is practically the same of the wind indices.

Finally, the intangible value of CH assets is considered in the definition of the prioritization scheme according to the procedure proposed in *Section 3.5*. In order to assess the validity of the proposed procedure the analysed CH assets are assumed to be characterised by local significance, except for the building 01-013, one of the assets which behave better, whose significance

is considered recognised at national level. Figure 9 shows the multi-hazard prioritization indices which consider the CH intangible value. The general trend is the same of the wind prioritization index, but the relative position of building 01-013 changes. This simple example shows that if the intangible value of CH assets within a given portfolio is not homogeneous it can drive the prioritization scheme.





**Figure 8: Prioritization indices: a) Seismic risk prioritization index; b) Wind risk prioritization index; c) Multi-hazard risk prioritization index. Background map by ©OpenStreetMap contributors 2019. Distributed under a Creative Commons BY-SA License.**






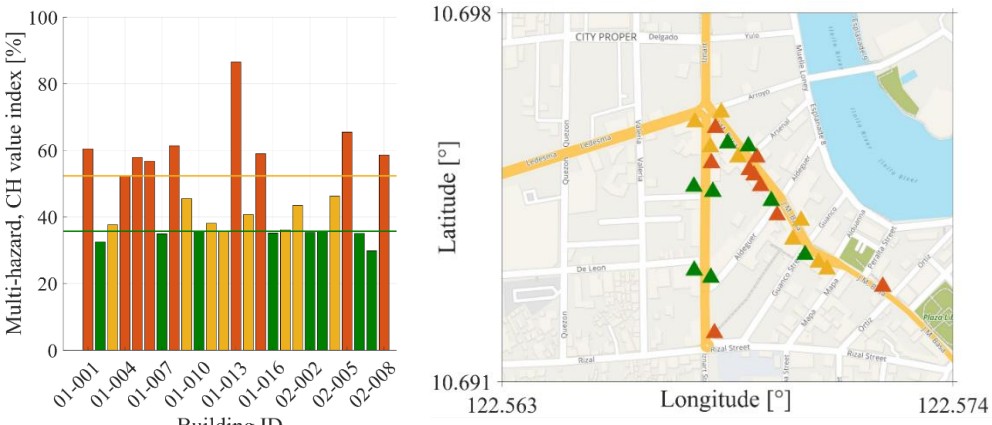

**Figure 9: Multi-hazard risk prioritization index which considers the CH intangible value. Background map by ©OpenStreetMap contributors 2019. Distributed under a Creative Commons BY-SA License.**

## 5 Concluding remarks

This paper presented a multi-hazard risk prioritization framework for CH assets which represents the lowest refinement level
of a multi-level risk and resilience assessment procedure. This procedure is indeed one of the first outcomes of the *Cultural Heritage Resilience & Sustainability to multiple Hazards* (CHeRiSH) project, which aims to develop a multi-level, harmonized, and engineering-based risk and resilience assessment framework for CH assets in the Philippines exposed to multiple natural hazards.

To this aim, an ad-hoc RVS form designed for CH assets has been introduced in this paper. In particular, the multi-level
architecture of the proposed RVS form allows one to improve the estimation of the structural fragility and risk once new detailed information is available. At the lowest refinement level (the main focus of the paper), the data gathered in the RVS form are used for the calculation of the proposed seismic and wind prioritization indices. They represent empirical proxies for the relative risk of CH assets within the analysed portfolio and then they can be used only for prioritization purposes.

The proposed seismic risk prioritization index extended the one developed within the INSPIRE project to the case of URM
buildings. It consists of two parts: a baseline score and a performance modifier. The baseline score calculation is based on the HAZUS model fragility curves, while the performance modifier is computed as weighted summation of scores related to macro-categories and secondary parameters, which, if present, are deemed to jeopardise the building performance. The macro-categories express the seismic failure chain peculiar of URM buildings. Each of them contributes to the calculation of the performance modifier through secondary parameters which express specific structural features which can prevent or promote
the activation of failure mechanisms, as observed during post-earthquake surveys. The proposed wind risk prioritization index was similarly defined as the weighted summation of scores and weights related to vulnerability factors of CH asset roofs multiplied by a hazard parameter. The vulnerability factors defined within the SCOSSO project have been adapted in this work to the needs of CH assets. A simple method to combine risk prioritization indices related to different hazards and which allows



considering the intangible value of CH assets has been finally introduced. The multi-hazard risk prioritization index was
calculated as the Euclidian norm of the vector whose components are the single-hazard prioritization indices. The intangible
CH asset value was considered by multiplying the multi-hazard risk prioritization index by a score that account for the
significance of the asset as CH. The Analytic Hierarchy Process (AHP) has been extensively used to calibrate combination
weights and scores, thus reducing the subjectivity involved in the procedure.

The application of the proposed prioritization framework on the CH assets of Iloilo City, Philippines, has shown its feasibility
in practice. Findings from the fieldwork highlight the important role played by the widespread vulnerability factors, strongly
affecting the performance of the surveyed CH assets. The case study highlighted the need of considering the intangible value
of CH assets within prioritization procedures.

This study represents a first step toward a comprehensive framework for multi-hazard risk assessment and optimal resilience-
enhancing strategy selection for CH assets. Future developments will aim to improve the quantification of the wind
vulnerability through the definition of suitable numerical models which consider degradation effects and climate change
impact.

### Acknowledgements

This study was performed in the framework of the "*CHeRiSH: Cultural Heritage Resilience & Sustainability to multiple
Hazards*" project funded by the UK British Council. The University of the Philippines Visayas (UPV), the Central Philippine
University (CPU), and the De La Salle University (DLSU), Manila, are acknowledged for the technical support during the
fieldwork. The graduate students from CPU, Mr. Kieran O'Sullivan from University College London and Miss. Sabrina di
Stasio from University of Naples Federico II (Italy), who participated in the fieldwork are also gratefully acknowledged.

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





## Appendix 1

**Table A.1: Judgment matrix adopted for the calibration of the macro-category weights.**

|  | Material quality | Local behaviour | Global behaviour | Façade ornaments |
|---|---|---|---|---|
| Material quality | 1 | 2 | 3 | 5 |
| Local behaviour | 1/2 | 1 | 2 | 5 |
| Global behaviour | 1/3 | 1/2 | 1 | 5 |
| Façade ornaments | 1/5 | 1/5 | 1/5 | 1 |

**Table A.2: Judgment matrix adopted for the calibration of the local behaviour weights.**

|  | Opening Layout | Wall Slenderness | Façade Regularity | Opening Area | Wall-to-Wall Connection | Wall-to-Diaphragm Connection | Wall-to-Roof Connection |
|---|---|---|---|---|---|---|---|
| Opening Layout | 1 | 2 | 1/2 | 1 | 1/3 | 1/6 | 1/3 |
| Wall Slenderness | 1/2 | 1 | 1/2 | 1/2 | 1/6 | 1/8 | 1/6 |
| Façade Regularity | 2 | 2 | 1 | 2 | 1/2 | 1/3 | 1/2 |
| Opening Area | 1 | 2 | 1/2 | 1 | 1/6 | 1/8 | 1/6 |
| Wall-to-Wall Connection | 3 | 6 | 2 | 6 | 1 | 1/3 | 1 |
| Wall-to-Diaphragm Connection | 6 | 8 | 3 | 8 | 3 | 1 | 2 |
| Wall-to-Roof Connection | 3 | 6 | 2 | 6 | 1 | 1/2 | 1 |

**Table A.3: Judgment matrix adopted for the calibration of the global behaviour weights.**

|  | Plan shape | Storey height uniformity | Added storeys | Pounding | Unfavourable soil |
|---|---|---|---|---|---|
| Plan shape | 1 | 2 | 2 | 1/2 | 1/2 |
| Storey height uniformity | 1/2 | 1 | 1 | 1/4 | 1 |
| Added storeys | 1/2 | 1 | 1 | 1/3 | 1/2 |
| Pounding | 2 | 4 | 3 | 1 | 4 |
| Unfavourable soil | 2 | 1 | 2 | 1/4 | 1 |

**Table A.4: Judgment matrix adopted for the calibration of the RC building weights.**

|  | Preservation condition | Plan Shape | Storey Height Uniformity | Added Storeys | Infills at ground storey | Short column | Pounding | Unfavourable Soil |
|---|---|---|---|---|---|---|---|---|
| Preservation condition | 1 | 1 | 2 | 2 | 1/3 | 1/2 | 1/2 | 2 |
| Plan shape | 1 | 1 | 2 | 2 | 1/3 | 1/2 | 1/2 | 1/2 |
| Storey height uniformity | 1/2 | 1/2 | 1 | 1 | 1/6 | 1/4 | 1/4 | 1 |




| | | | | | | | | |
|---|---|---|---|---|---|---|---|---|
| Added storeys | 1/2 | 1/2 | 1 | 1 | 1/6 | 1/4 | 1/4 | 1 |
| Infills at ground storey | 3 | 3 | 6 | 6 | 1 | 2 | 2 | 6 |
| Short column | 2 | 2 | 4 | 4 | 1/2 | 1 | 1 | 4 |
| Pounding | 2 | 2 | 4 | 4 | 1/2 | 1 | 1 | 4 |
| Unfavourable soil | 1/2 | 2 | 1 | 1 | 1/6 | 1/4 | 1/4 | 1 |

**Table A.5: Judgment matrix adopted for the calibration of the roof vulnerability factor weights.**

| | Code level | Number of storeys | Roof Structure | Roof Covering | Roof Pitch | Roof Condition | Roof Connection | Structural Condition |
|---|---|---|---|---|---|---|---|---|
| Code level | 1 | 3 | 2 | 2 | 2 | 1 | 1 | 1 |
| Number of storeys | 1/3 | 1 | 1/2 | 1/2 | 1/2 | 1/4 | 1/4 | 1/4 |
| Roof structure | 1/2 | 2 | 1 | 1 | 1 | 1/2 | 1/2 | 1/2 |
| Roof covering | 1/2 | 2 | 1 | 1 | 1 | 1/4 | 1/4 | 1/4 |
| Roof pitch | 1/2 | 2 | 1 | 1 | 1 | 1/2 | 1/2 | 1 |
| Roof condition | 1 | 4 | 2 | 4 | 2 | 1 | 1/2 | 1 |
| Roof connection | 1 | 4 | 2 | 4 | 2 | 2 | 1 | 1 |
| Structural condition | 1 | 4 | 2 | 4 | 1 | 1 | 1 | 1 |

**Table A.6: Judgment matrix adopted for the calibration of the hazard parameters.**

| | High wind hazard | Medium wind hazard | Low wind hazard |
|---|---|---|---|
| High wind hazard | 1 | 2 | 3 |
| Medium wind hazard | 1/2 | 1 | 2 |
| Low wind hazard | 1/3 | 1/2 | 1 |

**Table A.7: Judgment matrix adopted for the calibration of the CH value scores.**

| | Exceptional significance | Considerable significance | Some significance | Little significance |
|---|---|---|---|---|
| Exceptional significance | 1 | 2 | 3 | 4 |
| Considerable significance | 1/2 | 1 | 2 | 3 |
| Some significance | 1/3 | 1/2 | 1 | 2 |
| Little significance | 1/4 | 1/3 | 1/2 | 1 |