# Peer review of "A multi-hazard risk prioritisation framework for cultural heritage assets"

_Natural Hazards and Earth System Sciences, 2020_

## Referee Comment (RC1) · Anonymous Referee #1 · 24 Feb 2020

The manuscript focuses a very relevant topic and discusses interesting new ideas for prioritizing risk reduction measures in historic buildings. The manuscript is very clear and well written, and I only have the following minor comments/suggestions to make.

1) Regarding the Introduction, authors should clearly define the typology of cultural heritage assets that are focussed by the risk prioritization approach they propose. Since the referred CH assets are building-like reinforced concrete frame and masonry structures, the applicability of the proposed approach is restricted to this type of assets and this should be mentioned in the Introduction. This way the discussion in Section 2 will be framed more clearly. In the current version of the manuscript, the reader only finds this clearly stated in Section 3.1.

2) Still on the typology of the cultural heritage assets, authors refer that Filipino cultural

heritage also includes mixed structures. Reference to this type of structure is only mentioned in Section 3.1. Assuming that the proposed risk prioritization approach does not cover this type of structure (the presentation of the proposed approach only focuses structures that are either reinforced concrete or masonry), authors should refer this issue in Section 3.1

3) In line 2015 (page 8), authors discuss the definition of cultural heritage and refer to the criteria set by UNESCO in the Operational Guidelines for the Implementation of the World Heritage Convention. However, the line of reasoning suggested by the authors generalizes the context of the guidelines in a way that is not intended by their scope: the Operational Guidelines do not define cultural heritage, the Operational Guidelines define what is considered World Heritage, which within the UNESCO jargon is made of three types of heritage: natural, cultural and mixed (cultural and natural). As such, given the context that authors are discussing (i.e. different views of what is cultural heritage) perhaps a more suitable reference would be (Vecco, 2010)

Vecco, M. (2010). A definition of cultural heritage: From the tangible to the intangible. Journal of Cultural Heritage, 11(3), 321-324.

4) Figure 2 is a bit too small for clear readability. It could be included as a full-page figure (rotated 90°) or included as supplemental data. 5) In Section 3.5 authors discuss how to include the intangible nature of cultural heritage value within their approach. Although their proposal is adequate and simple, other approaches have been recently proposed that are able to provide a larger differentiation between the assets while maintaining an adequate level of simplicity (e.g. see the definition of baseline value in (Romão and Paupério, 2019)).

Romão, X., Paupério, E. (2019). An Indicator for Post-disaster Economic Loss Valuation of Impacts on Cultural Heritage. International Journal of Architectural Heritage, DOI: 10.1080/15583058.2019.1643948

2020-7, 2020.

---

## Referee Comment (RC2) · Fulvio Parisi (Referee) · 4 Apr 2020

The manuscript presents a very interesting multi-level procedure to prioritise disaster risk reduction measures for cultural heritage assets, considering multiple hazards. The manuscript is well organised and written, allowing a good understanding of the proposed framework. Just some typos can be detected in some instances, which can be easily removed. The methodology presented in this paper has different potentialities; for instance, the quantitative consideration of possible construction deficiencies at multiple scales is strongly appreciated because it may have a significant impact on relative risk estimates used in the prioritization scheme. Therefore, this reviewer recommends a minor revision of the manuscript according to the comments provided below.

[Figure]

1) The CHeRiSH RVS form illustrated in Fig. 2 includes the possible description of the "Opening Layout" ("Building Information" module), which may play a key role in the in-plane response of load-bearing URM walls to horizontal seismic actions. It seems that horizontal misalignment of openings at given storeys is not taken into account. Please comment on this and eventually include this feature in the form.

2) What is the meaning of "Frame masonry" and "Reinforced" in the section "Type of Lateral Load Resisting System" of the "Structural Information" module? Please make a double check of the taxonomy reported therein; it seems that the "Moment Resisting Frame System" type is not mentioned.

3) Did the authors evaluate the possibility of including adobe masonry in the section "Masonry Type" of the "Masonry" module? Regardless of the actual use of adobe masonry in the Philippines, the RVS form could include it to allow the implementation of CHeRiSH procedure in other countries.

4) Line 384: I suggest replacing "Wall-to-Diaphragm connection" by "Floor-to-Wall connection" because existing floors, particularly in old masonry buildings, do not necessarily develop a diaphragmatic action in the global seismic response. This also applies, for instance, in Fig. 4.

5) Table 3: It appears that the façade regularity depends on the opening layout, but they are separately scored. Is there any overlapping between scores? Please comment in the text.

---

## Author Comment (AC1) · 7 Apr 2020

**The manuscript focuses a very relevant topic and discusses interesting new ideas for prioritizing risk reduction measures in historic buildings. The manuscript is very clear and well written, and I only have the following minor comments/suggestions to make.**

We would like to thank this reviewer for the positive overall assessment of our contribution and for the insightful comments on our manuscript. Based on these comments, various revisions have been made to further improve the quality of the paper.

[Figure]

**1) Regarding the Introduction, authors should clearly define the typology of cultural heritage assets that are focussed by the risk prioritization approach they propose. Since the referred CH assets are building-like reinforced concrete frame and masonry structures, the applicability of the proposed approach is restricted to this type of assets and this should be mentioned in the Introduction. This way the discussion in Section 2 will be framed more clearly. In the current version of the manuscript, the reader only finds this clearly stated in Section 3.1.**

The proposed procedure for multi-hazard risk prioritization of CH assets focuses on reinforced concrete frames and unreinforced masonry buildings. We agree with this reviewer that specifying – from the introduction – the structural typologies considered in the proposed procedure can improve the discussion. In the revised version of the manuscript the following sentence has been added at line 71 (page 3):

*"...This paper addresses the above-mentioned issues by proposing a multi-level, multi-hazard risk assessment framework for CH assets , with a special focus on reinforced concrete (RC) frames and unreinforced masonry (URM) buildings..."*

**2) Still on the typology of the cultural heritage assets, authors refer that Filipino cultural heritage also includes mixed structures. Reference to this type of structure is only mentioned in Section 3.1. Assuming that the proposed risk prioritization approach does not cover this type of structure (the presentation of the proposed approach only focuses structures that are either reinforced concrete or masonry), authors should refer this issue in Section 3.1**

We agree with this reviewer. The modification of the sentence at line 71 (previous comment) already provides a clarification concerning the structural typologies considered

in the proposed framework. However, the following sentence has also been added at line 213 (page 8) of the revised manuscript to further clarify this aspect:

*"The proposed RVS form has been designed in order to account for the specific features of Filipino CH assets, which mainly consist of RC frames and masonry or mixed structures. It is worth noting, however, that even though the RVS form can be used to collect data related to combined structural typologies, they are not explicitly considered (in terms of scores and weights) in the proposed multi-hazard risk prioritization framework..."*

**3) In line 2015 (page 8), authors discuss the definition of cultural heritage and refer to the criteria set by UNESCO in the Operational Guidelines for the Implementation of the World Heritage Convention. However, the line of reasoning suggested by the authors generalizes the context of the guidelines in a way that is not intended by their scope: the Operational Guidelines do not define cultural heritage, the Operational Guidelines define what is considered World Heritage, which within the UNESCO jargon is made of three types of heritage: natural, cultural and mixed (cultural and natural). As such, given the context that authors are discussing (i.e. different views of what is cultural heritage) perhaps a more suitable reference would be (Vecco, 2010): Vecco, M. (2010). A definition of cultural heritage: From the tangible to the intangible. Journal of Cultural Heritage, 11(3), 321-324.**

We thank this reviewer for this reference which has been added to revised manuscript replacing the incorrect one.

**4) Figure 2 is a bit too small for clear readability. It could be included as a full-page figure (rotated 90) or included as supplemental data.**

[Figure]

We agree with this reviewer. While we have left figure 2 in the main text (Section 3.1) to provide the reader with a comprehensive view of the form (when first introduced), we will also add the full form (at its original 3 x A4-size page) as supplemental (online) material.

**5) In Section 3.5 authors discuss how to include the intangible nature of cultural heritage value within their approach. Although their proposal is adequate and simple, other approaches have been recently proposed that are able to provide a larger differentiation between the assets while maintaining an adequate level of simplicity (e.g. see the definition of baseline value in (Romão and Paupério, 2019)). Romão, X., Paupério, E. (2019). An Indicator for Post-disaster Economic Loss Valuation of Impacts on Cultural Heritage. International Journal of Architectural Heritage, DOI: 10.1080/15583058.2019.1643948 2020-7, 2020.**

We thank this reviewer for the reference, which has been added to the literature review (Section 2) at line 176 (page 6).

*. . .Romão and Paupério (2020) presented an approach for the quantification of economic losses related to CH assets damaged by catastrophic natural events. Particularly interesting, for the scope of this study, is the definition of the baseline pre-disaster value of the CH asset which namely corresponds to the asset intangible value. The authors consider four categories (i.e., evidential, historical, aesthetic and communal values) reflecting different levels of CH asset significance (Kerr, 2013). This approach requires only few information about the assets under investigation and then it can be used at portfolio level prioritization/assessment. . .*

Moreover, the following sentence has been added in Section 3.5 (line 517, page 21), where the proposed index is introduced.

*"... It is worth noting that the classification of the CH asset significance proposed by Kerr (2013) has been already successfully used/validated in the scientific literature for the quantification of the intangible value (e.g., Romão and Paupério, 2020; Figueiredo et al., 2019). This further strengthens the validity of the proposed procedure..."*

---

## Author Comment (AC2) · 7 Apr 2020

**The manuscript presents a very interesting multi-level procedure to prioritise disaster risk reduction measures for cultural heritage assets, considering multiple hazards. The manuscript is well organised and written, allowing a good understanding of the proposed framework. Just some typos can be detected in some instances, which can be easily removed. The methodology presented in this paper has different potentialities; for instance, the quantitative consideration of possible construction deficiencies at multiple scales is strongly appreciated because it may have a significant impact on relative risk estimates used in the prioritization scheme. Therefore, this reviewer recommends a minor revision of the manuscript according to the comments provided below.**

We sincerely thank Dr Parisi for the positive overall assessment of our contribution and for the insightful comments on our manuscript. Based on these comments, various revisions have been made to further improve the quality of the paper.

**1) Just some typos can be detected in some instances, which can be easily removed.**

We thank Dr Parisi for this observation. An additional proofreading of the manuscript has been performed to fix any typos and further improve the readability of the manuscript.

**2) The CHeRiSH RVS form illustrated in Fig. 2 includes the possible description of the "Opening Layout" ("Building Information" module), which may play a key role in the inplane response of load-bearing URM walls to horizontal seismic actions. It seems that horizontal misalignment of openings at given storeys is not taken into account. Please comment on this and eventually include this feature in the form.**

The in-plane behaviour of URM buildings is affected by the horizontal/vertical misalignment of openings as well as their layout (i.e., position on the façade, this is one of the factors determining the dimension of piers). The misalignment is considered through the entry "façade regularity" whose three options (i.e., 1. Regular, 2. Medium, 3. Irregular) express vertical/horizontal alignment of openings (Table 3). This way to parametrize the problem derives from D'Ayala and Speranza (2002)[1], which developed the FaMIVE (Failure Mechanism Identification and Vulnerability Evaluation) approach. We thank Dr Parisi for this comment as we understand this aspect was particularly

clear in the original manuscript/form. We have now changed the entry name "façade regularity" to "opening alignment" both in the RVS form (Fig.2) and in Table 3. We have also modified/added the following sentences to the revised version of the manuscript:

Line 392, page 15:
*"...Indeed, it is well known that the activation of out-of-plane local mechanisms is strictly linked to the geometry of the piers (i.e., Opening Layout), which is also determined by the position of the openings (i.e., Opening Alignment), and the connection with orthogonal walls, diaphragms and roof (D'Ayala, 2005)..."*

Line 397, page 15
*"...The dimension of the piers, which is linked to the Opening Layout and the Opening Alignment, affect both the out-of-plane and the in-plane behaviours (e.g., Parisi and Augenti, 2013) of the URM building resisting members. However, in the proposed approach, these secondary parameters are considered only in the Local Behaviour component to avoid counting their effect twice..."*

[1] *D'Ayala, D. and Speranza, E. (2002) An integrated procedure for the assessment of seismic vulnerability of historic buildings. In:12th European Conference on Earthquake Engineering, paper no. 561.*

**3) What is the meaning of "Frame masonry" and "Reinforced" in the section "Type of Lateral Load Resisting System" of the "Structural Information" module? Please make a double check of the taxonomy reported therein; it seems that the "Moment Resisting Frame System" type is not mentioned.**

We thank Dr Parisi for this observation. There were various inconsistencies in the

form that have now been corrected, resulting in various changes in the taxonomy of the sections: "Material of Lateral Load Resisting System" and "Type of Lateral-load Resisting System". The changes are reported below:

• "Material of Lateral Load Resisting System": the options "reinforced masonry" and "confined masonry" have been added.

• "Type of Lateral-load Resisting System": the options "frame masonry", "confined masonry", "reinforced" and "dual system" have been removed. Whereas the entries: "Moment Resisting Frame System", "Load Bearing Walls" and "Combined" have been added.

**4) Did the authors evaluate the possibility of including adobe masonry in the section "Masonry Type" of the "Masonry" module? Regardless of the actual use of adobe masonry in the Philippines, the RVS form could include it to allow the implementation of CHeRiSH procedure in other countries.**

The option "Adobe bricks" has now been added in the section "Masonry type" of the proposed RVS form. However, specific criteria for the formulation of baseline scores and performance modifiers should be further investigated, in future studies, to adapt the proposed approach to the specific case of adobe masonry.

**5) Line 384: I suggest replacing "Wall-to-Diaphragm connection" by "Floor-to-Wall connection" because existing floors, particularly in old masonry buildings, do not necessarily develop a diaphragmatic action in the global seismic response. This also applies, for instance, in Fig. 4.**

We agree with the reviewer, the term floor-to-wall connection better expresses the idea of the constraining effect of the floor to the piers. The term *Wall-to-Diaphragm*

[Figure]

*connection* has been now changed to *Floor-to-Wall connection* in the revised version of the manuscript/form.

**6) Table 3: It appears that the façade regularity depends on the opening layout, but they are separately scored. Is there any overlapping between scores? Please comment in the text.**

The answer to this comment is strictly related to the answer to the first comment above, and we thank again Dr Parisi for highlighting that this aspect was not very clear in the original manuscript. The criterion Façade regularity (which in the revised version of the manuscript is called Opening alignment) refers to the alignment of the openings, while the criterion Opening layout refers to the position of the opening on the façade. This latter is key to determine the pier geometry. We believe that the sentences added with reference to the first comment above also address the issue raised here, clarifying this aspect of the proposed approach.

---

## Author Response (AR1)

[revised manuscript text omitted]

**CHeRISH — RAPID VISUAL SURVEY**
Cultural Heritage Resilience & Sustainability to multiple Hazards

**GENERAL INFORMATION**

| | | |
|---|---|---|
| Date and Time: | Surveyor Name: | Nearby Buildings: [ ] Smaller [ ] Same Height [ ] Taller |
| Address: | | |
| No. of Building Users: | GPS Co-Ordinates: | |
| Construction Year: | Confidence: | [ ] H [ ] M [ ] L |

| Shape and Composition of the Block: | [ ] Triangular Shape-Synchronic Growth  [ ] Triangular Shape-Diachronic Growth  [ ] Bulk Shape-Synchronic Growth  [ ] Bulk Shape-Diachronic Growth | [ ] Elongated Shape-Synchronic Growth  [ ] Elongated Shape-Diachronic Growth  [ ] Individual Buildings |
|---|---|---|
| Position in Block: | [ ] Corner  [ ] Mid-block  [ ] End-block  [ ] Isolated  [ ] Other: | |
| Type of Survey: | [ ] Desktop Review  [ ] Exterior  [ ] Part. Interior  [ ] Interior | |

**BUILDING INFORMATION**

| | |
|---|---|
| No. of Stories: | |
| Storey Height (m): | |
| Average Height of Upper Horizontal Spandrel (m): | |
| Connection of the Walls at the Edges (Exterior): | [ ] Adequate  [ ] Poor |
| Wall Openings Max. Dim. (m × m): | |
| Wall Openings Total Area (m²): | |
| Opening Layout: | [ ] Opening with Vertical Alignment at Both the Edges of Façade  [ ] Openings with Vertical Alignment at an Edge of the Façade  [ ] Central Column of Opening, Vertically Aligned |
| Opening Alignment: | [ ] Regular [ ] Medium [ ] Irregular |
| Dim. Between Int. Structural Wall (m × m): | |

Simple Building Plan

| | | |
|---|---|---|
| Max. Thickness Ext. Walls (m): | | Min. Thickness Ext. Walls (m): |
| Max. Thickness Int. Walls (m): | | Min. Thickness Int. Walls (m): |
| Non-Continuous Structural Wall: | [ ] Yes [ ] No → Position: | |
| Plan Regularity: | [ ] Regular [ ] Medium [ ] Irregular Confidence: [ ] H [ ] M [ ] L | |
| Height Regularity: | [ ] Regular [ ] Medium [ ] Irregular Confidence: [ ] H [ ] M [ ] L | |
| Drawings: | [ ] Yes [ ] No [ ] Structural [ ] Architectural → File Name: | |

**ROOF INFORMATION**

| | | Unk | Confidence |
|---|---|---|---|
| Type: | [ ] Flat [ ] Mono Pitch [ ] Multi Pitch [ ] Gable | [ ] | [ ] H [ ] M [ ] L |
| Truss Material: | [ ] RC Slab [ ] Timber [ ] Steel [ ] Other | [ ] | [ ] H [ ] M [ ] L |
| Slope (°): | Soffit Width (m): | Mean Roof Height (m): | [ ] | [ ] H [ ] M [ ] L |
| Panel Material: | [ ] Timber [ ] Steel [ ] Other | Thickness (mm): | [ ] | [ ] H [ ] M [ ] L |
| Fastener Type: | [ ] Screw [ ] Nail [ ] Other | | [ ] | [ ] H [ ] M [ ] L |
| No. of Purlins: | No. of Fasteners Per Purlin Bay: | | [ ] | [ ] H [ ] M [ ] L |

©2019 EPICentre

| | | Unk | Confidence |
|---|---|---|---|
| Fastener Dia. (mm): | Fastener Penetration (mm): | [ ] | H M L |
| Roof-Wall Connection: | [ ] Simply Supported [ ] Pinned [ ] Fixed Support | [ ] | H M L |
| Roof-Wall Fastener: | [ ] Metal Plate Connector [ ] Single Hurricane Tie  [ ] Toe Nails [ ] Double Hurricane Tie | [ ] | H M L |
| Ornaments Type: | Material: | Dimension (m × m): | [ ] | H M L |

**STRUCTURAL INFORMATION**

| | | Unk | Confidence |
|---|---|---|---|
| Material of Lateral Resisting System: | [ ] Reinforced Concrete [ ] Masonry [ ] Timber  [ ] Steel [ ] Other: | [ ] | H M L |
| Type of Lateral Load Resisting System: | [ ] Frame Masonry [ ] Dual System [ ] Shear Wall  [ ] Confined Masonry [ ] Bracing  [ ] Reinforced [ ] Other: | [ ] | H M L |
| Structural Condition: | [ ] Poor / Deteriorated [ ] Good / Fair [ ] Excellent / New | [ ] | H M L |
| Environmental Exposure: | [ ] Dry Environment [ ] Aggressive Chemical Environment  [ ] Moisture or Wetting [ ] Saturated Salt Air | [ ] | H M L |
| Foundation Type: | [ ] Deep [ ] Superficial [ ] Not Accessible Note: | [ ] | H M L |
| Diaphragm Type: | [ ] Timber [ ] Concrete | [ ] | H M L |
| Load Distribution: | [ ] One-Way Spanning [ ] Two-Way Spanning | [ ] | H M L |
| Diaphragm-Wall Connection: | [ ] Simply Supported [ ] Steel Bars [ ] RC Ring Beam | [ ] | H M L |
| Retrofitting: | [ ] Yes [ ] No Description: | [ ] | H M L |
| Modifications: | [ ] Yes [ ] No → Position:  [ ] Addition of Stories [ ] Extension of Plan  [ ] Wall Opening Framing  [ ] Steel Frame Opening → Position: | [ ] | H M L |
| Vulnerability Factors: | [ ] Balconies [ ] Short Column  [ ] Parapet [ ] Strong Beam-Weak Column  [ ] Gable [ ] Soft Storey  [ ] Pounding [ ] Roof Thrust → Length x Height (m):  [ ] Mass Irregularity [ ] Vaults / arches → Length x Height (m):  [ ] Built on Slope [ ] Connection Between Orthogonal Wall (Interior): H M L  [ ] Built on Stilts [ ] Existing Cracks → Info:  [ ] Other: | | |

**MASONRY**

| | | Unk | Confidence |
|---|---|---|---|
| Masonry Type: | [ ] Chaotic Stone [ ] Masonry with Hewn Blocks  [ ] Hollow Brick [ ] Regular Sized Stone  [ ] Soft Stone Block  [ ] Squared Stone Blocks  [ ] Solid Brick Masonry and Lime Mortar  [ ] Hollow Brick with Cement Mortar  [ ] Hollow Brick without Mortar in Vertical Joints  [ ] Concrete Blocks or Expanded Clay Blocks  [ ] Concrete Hollow Blocks | [ ] | H M L |
| Mortar Type & Thickness: | [ ] Cement [ ] Mud  [ ] Mud with Cement [ ] Lime  [ ] Lime with Bricks [ ] Other: | Thickness (mm): | [ ] | H M L |

©2019 EPICentre
Prepared Kieran O'Sullivan & Giacomo Sevieri
* * *
260

[revised manuscript text omitted]